# Fast Rerandomization for Balancing Covariate in Randomized Experiments: A Metropolis–Hastings Framework

## Abstract

Balancing covariates is critical for credible and efficient randomized experiments. Rerandomization addresses this by repeatedly generating treatment assignments until covariate balance meets a prespecified threshold. By shrinking this threshold, it can achieve arbitrarily strong balance, with established results guaranteeing optimal estimation and valid inference in both finite-sample and asymptotic settings across diverse complex experimental settings. Despite its rigorous theoretical foundations, practical use is limited by the extreme inefficiency of rejection sampling, which becomes prohibitively slow under small thresholds and often forces practitioners to adopt suboptimal settings, leading to degraded performance. Existing work focusing on acceleration typically fail to maintain the uniformity over the acceptable assignment space, thus losing the theoretical grounds of classical rerandomization. Building upon a Metropolis-Hastings framework, we address this challenge by introducing an additional sampling-importance resampling step, which restores uniformity and preserves statistical guarantees. Our proposed algorithm, PSRSRR, achieves speedups ranging from 10 to 10,000 times while maintaining exact and asymptotic validity, as demonstrated by simulations and two real-data applications.

## 1 Introduction

Randomized experiments are the gold standard for credible causal inference as random assignment balances both observed and unobserved confounders in expectation. However, in practice, even under complete randomization, there remains a nontrivial risk of covariate imbalance (Rosenberger & Sverdlov, 2008), which grows as the number of covariates increases (Krieger et al., 2019; Morgan & Rubin, 2012). Such imbalance can reduce credibility due to accidental bias. While deterministic allocation can enforce near-exact covariate balance, it introduces its own problems, including selection bias, a loss of robustness, and the inability to use randomization-based inference (Harshaw et al., 2024).

An intuitive and practical approach to achieve what Kapelner et al. (2021) describe as "a harmony of optimal deterministic design and completely randomized design" is to randomize repeatedly until an assignment with appropriate and satisfactory covariate balance is achieved, a procedure known as rerandomization. Despite its long history and widespread use (Student, 1938; Cox, 1982; Bailey & Rowley, 1987; Maclure et al., 2006; Imai et al., 2008; Bruhn & McKenzie, 2009), the theoretical implications of rerandomization were first formally studied by Morgan & Rubin (2012) using the Mahalanobis distance. Since then, rerandomization has attracted growing interest, and its theoretical foundations have been established across various scenarios (Morgan & Rubin, 2015; Li et al., 2018; Zhou et al., 2018; Li et al., 2020; Shi et al., 2024; Wang et al., 2023b; Lu et al., 2023).

Although rerandomization can, in theory, achieve asymptotically optimal precision by shrinking the balance threshold as the sample size grows (Wang & Li, 2022), it is often regarded as *computationally infeasible* and therefore implemented with suboptimal thresholds when compared with alternative methods (Yang et al., 2023a; Harshaw et al., 2024). Because the statistical properties of rerandomization critically depend on the choice of threshold, this computational issue leads to suboptimal statistical performance and results in unfair comparisons with existing methods. The

computational limitation of rerandomization stems from its reliance on naive rejection-sampling, which typically yields a single accepted assignment only after evaluating thousands of candidate allocations based on their Mahalanobis distance. In practice, as the number of covariates increases, the optimal acceptance probability can become astronomically small (e.g., $< 10^{-15}$ with 20 covariates), making naive rejection–sampling implementations of rerandomization practically infeasible.

This computational challenge is amplified when applying Fisher randomization tests (FRT) under rerandomization. Constructing the null distribution requires generating hundreds or thousands of acceptable assignments, compounding an already slow and costly process. Yet, FRT is particularly vital in small-sample settings: Johansson et al. (2021) shows that the asymptotic confidence–interval results of Li et al. (2018) fail to control Type I error when $n$ is limited. Consequently, FRT has been widely advocated as a robust alternative to the asymptotic inference in Li et al. (2018), supported by extensive empirical evidence (Bind & Rubin, 2020; Keele, 2015; Proschan & Dodd, 2019; Young, 2019) and theoretical analyses (Branson, 2021; Cohen & Fogarty, 2022; Caughey et al., 2023; Luo et al., 2021; Wu & Ding, 2021; Zhao & Ding, 2021).

Several methods have been proposed to address the pressing need for accelerating rerandomization, including incorporating heuristic rules to search for satisfactory assignments (Krieger et al., 2019; Zhu & Liu, 2023), directly tackling the tradeoff between robustness and covariate balance by a Gram-Schmidt design (Harshaw et al., 2024), and engineering techniques such as via key-based storage and GPU/TPU backends (Goldstein et al., 2025). Among these approaches, the most relevant to our study is Zhu & Liu (2023), which uses pair-switching to search for a well-balanced allocation. This procedure substantially reduces computational cost and improves practical feasibility. However, due to the nature of the algorithm, it does not guarantee to sample uniformly from the set of acceptable assignments. Consequently, the uniformity condition underpinning asymptotic randomization-based inference is not assured, and those established theoretical results for classical rerandomization in Li et al. (2018) and Wang & Li (2022) do not directly apply.

Our contributions are twofold—theoretical and practical. On the theoretical side, we build on the Metropolis–Hastings framework to construct a Markov chain over the space of treatment assignments via pair switching. We derive the stationary distribution of this Markov chain, thus establishing the distribution of the acceptable assignment space it generates. Starting from this stationary distribution, we then apply rejection sampling to restore uniformity over this acceptable space. These results provide formal guarantees for the uniformity of the generated assignments, therefore validating theoretical results derived for classical rerandomization. Because the guarantees are asymptotic (valid as the number of iterations increases), we introduce an efficient stopping rule and translate our framework into a practical algorithm, PSRSRR. We show on both simulated and real data that PSRSRR yields assignment sets that are approximately uniformly distributed while substantially reducing sampling time compared to classical rerandomization. Our method bridges theory and practice, delivering a fast and reliable rerandomization procedure with desired theoretical guarantees.

The remainder of this paper is organized as follows. In Section 2, we introduce some preliminary knowledge of classical rerandomization. In Section 3, we develop our methodology in three steps: We begin with the formulation of the Markov chain given by pair-switching assignments, and derive the theoretical results on the generated distribution. We then incorporate rejection sampling to restore uniformity with theoretical foundations. Finally, we turn this theoretical result into a practical algorithm with a stopping rule that accelerates the rerandomization process while maintaining good enough uniformity. In Section 4, we present comprehensive experiment results on simulated and real data to illustrate the superior performance of our proposed algorithm in both estimation efficiency and sampling speed. We summarize and discuss our results in Section 5. Technical proofs and additional experimental results are in Appendix A and B. In Appendix C, we discuss extended related work on carefully designed randomized experiments in causal ML, A/B testing, and treatment effect estimation, with a focus on how rerandomization provides a principled way to improve covariate balance across various scenarios.

## 2 PRELIMINARIES

### 2.1 THE NEYMAN-RUBIN POTENTIAL OUTCOME FRAMEWORK

This study adopts the Neyman-Rubin potential outcome framework (Neyman, 1923; Rubin, 1974). We consider an experiment with $n$ units randomly drawn from a population, among which $n_t$ units are treated and $n_c = n - n_t$ units are controlled. We assume $n_t \geq 2$ and $n_c \geq 2$. We denote $\mathbf{W} = (W_1, \ldots, W_n)^\top$ as the vector of treatment assignment indicators, where $W_i = 1$ if unit $i$ receives treatment and $W_i = 0$ otherwise. For each unit $i$, we consider the existence of two potential outcomes, $(Y_i(1), Y_i(0))$, and assume the stable unit treatment value assumption (SUTVA) (Rubin, 1980), i.e., $Y_i = W_i Y_i(1) + (1 - W_i) Y_i(0)$. The unit-level treatment effect for unit $i$ is defined as $\tau_i = Y_i(1) - Y_i(0)$, and the average treatment effect is defined as $\tau = \frac{1}{n} \sum_{i=1}^{n} (Y_i(1) - Y_i(0))$, which could be estimated using the difference-in-means estimator $\widehat{\tau}(\mathbf{W}) = \frac{1}{n_t} \sum_{i:W_i=1} Y_i(1) - \frac{1}{n_c} \sum_{i:W_i=0} Y_i(0)$. For each unit, we also observe $p$ baseline covariates, denoted by $\mathbf{X}_i = (X_{i1}, \ldots, X_{ip})^\top$. The covariates of all units are gathered in a matrix $\mathbf{X} = (\mathbf{X}_1, \ldots, \mathbf{X}_n)^\top$, and the corresponding covariance matrix is denoted by $\mathbf{S}_{XX} = \frac{1}{n-1} \sum_{i=1}^{n} (\mathbf{X}_i - \overline{\mathbf{X}})(\mathbf{X}_i - \overline{\mathbf{X}})^\top$, where $\overline{\mathbf{X}} = \frac{1}{n} \sum_{i=1}^{n} \mathbf{X}_i$.

### 2.2 CLASSICAL RERANDOMIZATION USING THE MAHALANOBIS DISTANCE

Mahalanobis distance can be used to measure the covariate balance between the treatment and control groups. For a given assignment $\mathbf{W}$, the Mahalanobis distance is defined as

$$M(\mathbf{W}) := \left(\overline{\mathbf{X}}_t - \overline{\mathbf{X}}_c\right)^\top \left[\text{Cov}\left(\overline{\mathbf{X}}_t - \overline{\mathbf{X}}_c\right)\right]^{-1} \left(\overline{\mathbf{X}}_t - \overline{\mathbf{X}}_c\right),$$

where $\overline{\mathbf{X}}_t = \frac{1}{n_t} \sum_{i:W_i=1} \mathbf{X}_i$, $\overline{\mathbf{X}}_c = \frac{1}{n_c} \sum_{i:W_i=0} \mathbf{X}_i$, and $\text{Cov}\left(\overline{\mathbf{X}}_t - \overline{\mathbf{X}}_c\right) = \frac{n}{n_t n_c} \mathbf{S}_{XX}$. Morgan & Rubin (2012) suggested performing randomization by sampling a treatment assignment $\mathbf{W}$ from the set $\mathcal{W} = \{W \in \mathbb{R}^n : \sum_{i=1}^{n} W_i = n_t, W_i \in \{0, 1\}\}$. For the sampled assignment $\mathbf{W}$, its Mahalanobis distance $M(\mathbf{W})$ is compared against a pre-specified threshold $a$ that controls the acceptance level of the covariate imbalance. If $M(\mathbf{W}) \leq a$, the assignment is accepted; otherwise, the sampling process is repeated until a satisfactory assignment is found. We refer to rerandomization based on this acceptance-rejection sampling strategy as RR, and denote the set formed by all acceptable assignments as $\mathcal{W}_a = \{\mathbf{W} \in \mathcal{W} : M(\mathbf{W}) \leq a\}$.

When prior knowledge suggests that certain covariates are more strongly associated with potential outcomes than others, it is beneficial to adopt balance criteria that prioritize the more important covariates. This motivates the Bayesian criterion for rerandomization (Liu et al., 2025). In Section B.5, we demonstrate that our proposed framework can be readily adapted to this setting, offering a computationally efficient implementation for such advanced balance criteria.

## 3 METHOD

### 3.1 STATIONARY DISTRIBUTION OF PAIR-SWITCHING MARKOV CHAIN

Classical rerandomization inefficiently searches for balanced assignments via rejection sampling. To improve this process, we propose a constructive approach based on a Metropolis-Hastings algorithm. Our method starts with a single random assignment and iteratively refines it. In each step, a candidate assignment is proposed by swapping a randomly selected treatment-control pair. This candidate is then accepted or rejected based on a probability determined by the change in the Mahalanobis distance, $M(\mathbf{W})$. The temperature, $T$, is a tuning parameter that controls the likelihood of accepting a candidate with a worse balance (i.e., a higher $M(\mathbf{W})$), allowing the search to escape local minima. This process is repeated for a fixed number of iterations, $N$. The complete procedure is detailed in Algorithm 1, which will serve as a crucial building block for our final proposed algorithm PSRSRR.

While our proposed framework builds on the pair-switching strategy from Krieger et al. (2019) and Zhu & Liu (2023), a critical distinction lies in the stopping rule. Unlike Krieger et al. (2019), which terminates at a local optimum, or Zhu & Liu (2023), which stops immediately when $M(\mathbf{W}) \leq a$, our algorithm allows the chain to evolve toward its stationary distribution. Since previous methods

fail to reach a tractable stationary distribution, they cannot enable the subsequent recovery of a uniform distribution over $\mathcal{W}_a$.

---

**Algorithm 1:** Truncated Pair-Switching

---

**Input:** Covariates data $\mathbf{X}$, temperature $T$, max iteration number $N$.

Set $t = 0$;

Set $\mathbf{W}^{(0)}$ as $n_t$ elements equal to 1 and $n_c$ elements equal to 0 with random positions;

Set $M^{(0)} = M\left(\mathbf{W}^{(0)}\right)$;

**while** $t < N$ **do**

    Randomly switch the positions of one of the 1's and one of the 0's in $\mathbf{W}^{(t)}$ and obtain $\mathbf{W}^*$;

    Set $M^* = M\left(\mathbf{W}^*\right)$;

    Sample $J$ from a Bernoulli distribution with probability $\min\{\left(M^{(t)}/M^*\right)^{1/T}, 1\}$;

    **if** $J = 1$ **then**

        | Set $\mathbf{W}^{(t+1)} = \mathbf{W}^*$;

    **end**

    **else**

        | Set $\mathbf{W}^{(t+1)} = \mathbf{W}^{(t)}$;

    **end**

    Set $t = t + 1$;

**end**

**Output:** $\mathbf{W} = \mathbf{W}^{(N)}$.

---

By the definition of Markov chain (see for example, Givens & Hoeting (2012)), the sequence $\{\mathbf{W}^{(t)}\}_{t \geq 0}$ in Algorithm 1 forms a Markov chain over the space of all valid assignments, $\mathcal{W}$. As a result, as the number of iterations $N \to \infty$, the distribution of the generated assignments converges to a stationary distribution. This stationary distribution is characterized by the following theorem.

**Theorem 1** *The limiting distribution of the Markov chain $\{\mathbf{W}^{(t)}\}_{t \geq 0}$ with temperature $T$ is $\pi(\mathbf{W}) = \frac{M(\mathbf{W})^{-1/T}}{\sum_{\mathbf{W}^* \in \mathcal{W}} M(\mathbf{W}^*)^{-1/T}}$ for any $\mathbf{W} \in \mathcal{W}$. $\pi$ is also the stationary distribution.*

This stationary distribution is not uniform; the probability of sampling an assignment, $\pi(\mathbf{W})$, is inversely proportional to its Mahalanobis distance raised to a positive exponent ($\pi(W) \propto M(\mathbf{W})^{-1/T}$). We will leverage this non-uniform distribution in the next section to generate assignments that are uniform over the acceptable set, $\mathcal{W}_a$.

### 3.2 REJECTION SAMPLING

Although Algorithm 1 introduces a probabilistic mechanism for assignment generation and possesses a valuable theoretical guarantee, it can not be used directly for rerandomization because of two reasons. First, its final output is not guaranteed to be an acceptable assignment (i.e., it may have a Mahalanobis distance greater than the threshold). Second, its final output is not uniformly distributed. This departure from uniformity invalidates the theoretical guarantees that underpin classical rerandomization and can compromise the statistical efficiency of the resulting treatment effect estimates. Prior work like the PSRR method (Zhu & Liu, 2023) solves the first problem by letting the algorithm stop at the first accpetable assignment, but fails to solve the second.

To solve these challenges, we introduce a second step based on the principle of rejection sampling. The key insight is to treat the non-uniform stationary distribution $\pi$ (from Theorem 1) as a proposal distribution and then apply a corrective filter to obtain our target uniform distribution over the acceptable set $\mathcal{W}_a$. We achieve this based on a carefully designed formula of the acceptance probability.

The acceptance rule has two components. First, to ensure acceptability, any proposed assignment $\mathbf{W}$ that does not meet the balance criterion ($M(\mathbf{W}) > a$) is automatically rejected by setting its acceptance probability to zero. Second, to ensure uniformity among the remaining candidates, we apply the "inverse back" strategy. From Theorem 1, we know the probability of proposing an assignment is *inversely proportional to* $M(\mathbf{W})^{1/T}$. To cancel this known bias, the acceptance probability

for a valid candidate is made *directly proportional to* $M(\mathbf{W})^{1/T}$. The initial sampling bias and the corrective acceptance probability thereby cancel each other out, making the final probability constant for all assignments in $\mathcal{W}_a$. This two-part rule, formalized in Algorithm 2, results in a uniform sample from the acceptable set.

---

**Algorithm 2:** Rejection Sampling of Truncated Pair-Switching

---

**Input:** Covariates data $\mathbf{X}$, temperature $T$, max iteration number $N$, threshold $a$.
Set $Acc = \text{False}$;
**while** $Acc = \text{False}$ **do**
    Run Algorithm 1 with inputs$(\mathbf{X}, T, N)$ to generate assignment $\mathbf{W}$ ;
    Determine the acceptance probability

$$p(\mathbf{W}) = \begin{cases} (M(\mathbf{W})/a)^{1/T} & M(\mathbf{W}) \leq a \\ 0 & M(\mathbf{W}) > a \end{cases} \tag{1}$$

    Sample $J$ from Bernoulli variable with probability $p(\mathbf{W})$;
    **if** *J = 1* **then**
        | Set $Acc = \text{True}$;
    **end**
**end**
**Output:** $\mathbf{W}$.

---

**Theorem 2** *The assignment $\mathbf{W}$ generated by Algorithm 2 follows a uniform distribution on $\mathcal{W}_a$; that is, each $\mathbf{W} \in \mathcal{W}_a$ is selected with equal probability.*

We defer the proof of Theorem 2 to Appendix A. Since the assignments generated by Algorithm 2 follow the uniform distribution over $\mathcal{W}_a$, which is a fundamental assumption in Li et al. (2018), we can verify the unbiasedness of the resulting treatment effect estimator immediately and build upon their theoretical guarantees to construct asymptotic confidence intervals.

**Corollary 3** *Let $\chi_{p,a}^2 \sim \chi_p^2 \mid (\chi_p^2 \leq a)$ be a truncated $\chi^2$ random variable, $U_p$ be the first coordinate of the uniform random vector over the $(p-1)$-dimensional unit sphere. Let $\nu_\xi \left( R^2, p_a, p \right)$ be the $\xi$ th quantile of $\sqrt{1-R^2} \cdot \varepsilon_0 + \sqrt{R^2} \cdot \chi_{p,a} U_p$ where $\varepsilon_0 \sim \mathcal{N}(0,1)$. Denote by $\boldsymbol{s}_{Y(i),\boldsymbol{X}}$ the sample covariance between potential outcomes and covariates, $s_{Y(i)}^2$ the sample variance of potential outcomes, and $s_{Y(i)|\boldsymbol{X}}^2 = \boldsymbol{s}_{Y(i),\boldsymbol{X}} \mathbf{S}_{XX}^{-1} \boldsymbol{s}_{\boldsymbol{X},Y(i)}$ the sample variance of the linear projection of potential outcomes on covariates. Let $s_{\tau|\boldsymbol{X}}^2 = \left( s_{Y(1),\boldsymbol{X}} - s_{Y(0),\boldsymbol{X}} \right) \left( \boldsymbol{S}_{XX} \right)^{-1} \left( s_{\boldsymbol{X},Y(1)} - s_{\boldsymbol{X},Y(0)} \right)$. Let $\hat{V}_{\tau\tau} = n/n_t \cdot s_{Y(1)}^2 + n/n_c \cdot s_{Y(0)}^2 - s_{\tau|\boldsymbol{X}}^2$. Let $\hat{R}^2 = \hat{V}_{\tau\tau}^{-1} \left\{ n/n_t \cdot s_{Y(1)|\boldsymbol{X}}^2 + n/n_c \cdot s_{Y(0)|\boldsymbol{X}}^2 - s_{\tau|\boldsymbol{X}}^2 \right\}$. An asymptotic $(1-\alpha) \times 100\%$ confidence interval of the difference-in-means estimator is given by $\tau \in \left[ \hat{\tau} - \nu_{\alpha/2} \left( \hat{R}^2, p_a, p \right) \sqrt{\hat{V}_{\tau\tau}/n}, \quad \hat{\tau} - \nu_{1-\alpha/2} \left( \hat{R}^2, p_a, p \right) \sqrt{\hat{V}_{\tau\tau}/n} \right].$*

### 3.3 PRACTICAL IMPLEMENTATION

The procedure in Algorithm 2 is theoretically perfect: it is guaranteed to produce a uniform sample from the acceptable set $\mathcal{W}_a$. However, it can be computationally slow, as it requires running a full Markov chain (Algorithm 1) for many iterations just to generate a single candidate, which might then be rejected. This process is repeated until a candidate is finally accepted.

To bridge the gap between theoretical purity and practical speed, we introduce our main algorithm, Pair-Switching Rejection Sampling Rerandomization (PSRSRR). This algorithm fuses the Markov chain search and the rejection sampling check into a single, efficient procedure. Instead of running a full chain to draw a single candidate from the stationary distribution, we run a single chain and perform the acceptance check on-the-fly.

As the chain evolves from state $\mathbf{W}^{(t)}$ to $\mathbf{W}^{(t+1)}$, we check if the new state is acceptable (i.e., if $M(\mathbf{W}^{(t+1)}) < a$). If it is, we immediately apply the "inverse back" rejection sampling step. The chain terminates the very first time a candidate passes this second check. This practical procedure is formalized in Algorithm 3.

---

**Algorithm 3:** Pair-Switching Rejection Sampling Rerandomization (PSRSRR)

---

**Input:** Covariates data $\mathbf{X}$, threshold $a$, temperature $T$.

Set $Acc =$ False;

Set $t = 0$;

Set $\mathbf{W}^{(0)}$ as $n_t$ elements equal to 1 and $n_c$ elements equal to 0 with random positions;

Set $M^{(0)} = M\left(\mathbf{W}^{(0)}\right)$;

**while** $Acc =$ False **do**

    Randomly switch the positions of one of the 1's and one of the 0's in $\mathbf{W}^{(t)}$ and obtain $\mathbf{W}^*$;

    Set $M^* = M\left(\mathbf{W}^*\right)$;

    Sample $J$ from a Bernoulli distribution with probability $\min\left\{\left(M^{(t)}/M^*\right)^{1/T}, 1\right\}$;

    **if** $J = 1$ **then**

        Set $\mathbf{W}^{(t+1)} = \mathbf{W}^*$;

        Set $M^{(t+1)} = M^*$;

        **if** $M^{(t+1)} < a$ **then**

            Sample $\tilde{J}$ from a Bernoulli distribution with probability $\left(M^{(t+1)}/a\right)^{1/T}$;

        **end**

        **if** $\tilde{J} = 1$ **then**

            Set $Acc =$ True;

        **end**

    **end**

    **else**

        Set $\mathbf{W}^{(t+1)} = \mathbf{W}^{(t)}$;

    **end**

    Set $t = t + 1$;

**end**

**Output:** $\mathbf{W} = \mathbf{W}^{(t)}$.

---

As a first step toward a theoretical understanding of this practical implementation, we analyze a generalized framework in Appendix A.3. This framework modifies the algorithm by enforcing two explicit constraints: a *burn-in period* $L_{\mathrm{burn}}$, during which the chain evolves without performing rejection sampling to ensure convergence to stationarity; and a *sampling interval* $s_{\mathrm{chk}}$, such that the rejection sampling step is performed only every $s_{\mathrm{chk}}$ steps. We prove that if the burn-in is sufficiently long and the rejection sampling steps are sufficiently spaced, the output distribution of this generalized process converges to the target uniform distribution $\mathrm{Unif}(\mathcal{W}_a)$ in total variation distance.

Algorithm 3 corresponds to the limiting case of this framework where $L_{\mathrm{burn}} = 0$ and $s_{\mathrm{chk}} = 1$. While we do not provide a formal non-asymptotic bound for this specific parameter setting, we hypothesize that the algorithm relies on an "implicit burn-in" mechanism: since the acceptance probability is typically very small, the algorithm naturally runs for a large number of iterations before accepting a candidate. If this expected waiting time significantly exceeds the mixing time of the Markov chain, the distribution is expected to converge to stationarity before acceptance occurs, thereby mimicking the behavior of the explicit burn-in process. We leave the rigorous characterization of this implicit burn-in regime as a direction for future work.

# 4 EXPERIMENTS

## 4.1 SIMULATION STUDIES

**Objective**   Our simulation studies have three primary objectives. First, we empirically test whether our practical algorithm generates a nearly uniform distribution over the set of acceptable assignments. Second, we compare our method against competing methods on key statistical metrics, including mean squared error (MSE), confidence interval coverage, and statistical power. Third, we compare the computational time of our method with existing methods to demonstrate the computational efficiency of our method.

**Simulation setup**   Covariates are drawn from the standard normal distribution identically and independently: $X_{ij} \overset{i.i.d.}{\sim} N(0,1)$, $i = 1, \ldots, n$, $j = 1, \ldots, p$. The potential outcomes of the control group are generated independently from a linear model, $Y_i(0) = \sum_{j=1}^{p} X_{ij} + \epsilon_i$, where $\epsilon_i \sim N(0, \sigma^2)$. $\sigma^2$ is selected such that $R^2 = \mathbb{V}(\sum_{j=1}^{p} X_{ij})/\mathbb{V}(\sum_{j=1}^{p} X_{ij} + \epsilon_i) = 0.2, 0.5$ or $0.8$. We conduct simulations under both the null hypothesis and the alternative hypothesis, where we set $Y_i(1) = Y_i(0)$ and $Y_i(1) = Y_i(0) + 0.3\sqrt{\mathbb{V}[Y_i(0)]}$, respectively. The sizes of the treatment group and the control group are set as equal. More detailed settings regarding sample sizes $n$ and their corresponding sets of number of covariates $p$ are deferred to Appendix B.1.

We use two strategies for setting the acceptance threshold $a$. The first is the conventional approach, which sets $a$ to a small quantile of the $\chi_p^2$ distribution, (e.g., $p_a = \mathbb{P}\left(\chi_p^2 \leq a\right) = 10^{-3}$ or $10^{-5}$). The second strategy chooses $a$ based on the desired asymptotic variance reduction. Specifically, we set $\nu_{p,a} = \mathbb{P}\left(\chi_{p+2}^2 \leq a\right)/\mathbb{P}\left(\chi_p^2 \leq a\right) = 0.01$ as recommended by Wang & Li (2022).

We set the temperature hyperparameter $T$ using the empirical rule $T = 1.8/p$. The intuition is that as the number of covariates $p$ increases, the Mahalanobis distance landscape becomes smoother, meaning a single pair-switch yields a smaller change in $M(\mathbf{W})$. A lower temperature is therefore appropriate, as large, unfavorable jumps become less necessary to explore the assignment space effectively. A sensitivity analysis in Appendix B.2 shows that the algorithm is generally robust to temperature variations, although extremely low temperatures can drastically increase sampling time. Our empirical rule $T = 1.8/p$ ensures computational efficiency while maintaining favorable statistical performance.

**Competing methods**   We benchmark our method against four competing methods, using hyperparameters as recommended in their respective papers: (1) PSRR with temperature as $0.1$ and $p_a = \mathbb{P}\left(\chi_p^2 \leq a\right) = 10^{-5}$ (Zhu & Liu, 2023); (2) GSW with $\phi = 0.1$ (Harshaw et al., 2024). (3) Classical rerandomization (RR) with acceptance threshold $M(\mathbf{W}) < a$, where $a$ satisfies $p_a = \mathbb{P}\left(\chi_p^2 \leq a\right) = 10^{-3}$; (4) Complete randomization (CR) that randomly draws assignments from all possible ones.

**Uniformity**   We provide strong indirect evidence for the near-uniformity of our practical algorithm throughout our main results, as the validity of the statistical inference we construct relies on the fundamental assumption of uniformity. We defer two direct statistical evaluation tools using the Kolmogorov-Smirnov test to Appendix B.3.

**Evaluation metrics for estimation and inference**   For each method, we evaluate the performance of its resulting treatment effect estimator using five key statistical metrics. To assess estimation efficiency, we measure the estimator's mean squared error (MSE) and the average length of its corresponding confidence interval (CI). Both are reported as ratios relative to CR. To assess the validity of statistical inference, we evaluate the CI Coverage Rate (which should exceed the nominal 95% level), the Type I Error rate under the null hypothesis, and the statistical power under the alternative hypothesis.

**Comparison of estimation and inference results**   For each method, we sample 1000 assignments to evaluate their estimation and inference performance. The results for $R^2 = 0.5$ are presented below; additional reults can be found in Appendix B.4. As demonstrated by Figure 1, PSRSRR has the best and comparable performance as GSW in terms of the relative MSE, and the best performance

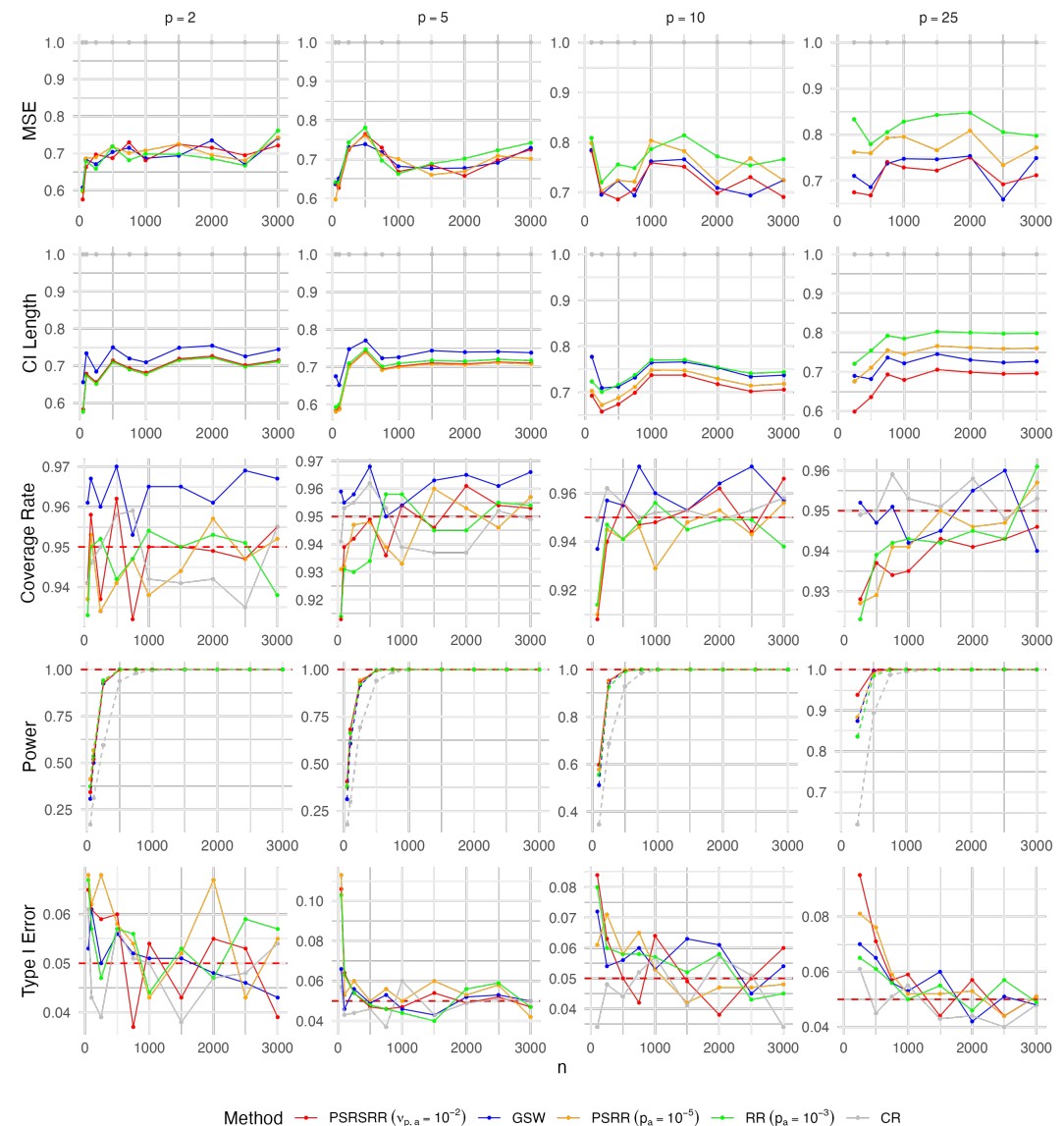

Figure 1: Comparison plots of, from top to bottom, MSE (relative to CR), CI length (relative to CR), coverage rate, power and type I error, by sample size $n$ and number of covariates $p$. The red dashed lines represent the nominal level of 95% CI coverage rate and 0.05 type I error. Better performance is indicated by lower MSE and CI Length, a coverage rate at or above 95%, higher power, and a type I error at or below 0.05.

in the relative confidence interval length. More detailed simulation results show that PSRSRR could have relative MSE ratio compared with CR as $58\% \sim 78\%$, $81\% \sim 107\%$ with RR, and $88\% \sim 107\%$ with PSRR. And when compared with PSRR, the improvement in MSE appears to be more significant when $p$ is larger, for example, when $p = 25$, the relative MSE ratio compared with PSRR is $88\% \sim 94\%$. And we can obtain similar statistics for the relative confidence interval length ratio, which is $58\% \sim 74\%$ compared with CR, $83\% \sim 101\%$ with RR, $89\% \sim 101\%$ with PSRR, and $86\% \sim 97\%$ with GSW. As for CI coverage rate, power, and Type I error, PSRSRR achieves satisfactory results and has comparable or better performance than the other competing methods.

**Comparison of sampling time**    We compare the computational efficiency of each method by measuring the average time required to generate 100 assignments. As shown in Figure 2, PSRSRR is highly efficient. Its sampling time is comparable to the fast but non-uniform PSRR method and is

faster than GSW and RR. Notably, PSRSRR achieves a dramatic speedup over these competitors, which is, on median, 4 times faster than GSW and over 1,800 times faster than RR.

Figure 2: Comparison of sampling time (on a $\log_{10}$ scale) by sample size $n$ and number of covariates $p$. CR is plotted to provide a baseline for the computational cost of a single random draw, rather than as a benchmark to outperform.

## 4.2 APPLICATION TO REAL DATASETS

We apply PSRSRR to two real-world experimental datasets, one is the reserpine data (Jones, 2017) with 30 participants, and the other is the data from the Student Achievement and Retention (STAR) Project (Angrist et al., 2009) with nearly 1000 participants. The difference in these two datasets helps us to examine the performance of algorithms in both small-sample and large-sample circumstances, thus providing a more overall illustration. We provide a description for the analysis of STAR data and defer that of reserpine data and other details to Appendix B.7.

Table 1: Sampling Time Comparison in Reserpine and STAR Data.

| Data | PSRSRR ($\nu_{p,a}$) | PSRSRR ($p_a$) | PSRR | RR | CR |
|---|---|---|---|---|---|
| Reserpine | – | 0.47s | 0.30s | 22.70s | 0.03s |
| STAR | 2.95s | 3.63s | 1.28s | 193.65s | 0.21s |

**STAR data**  Similar to the pre-processing procedures in Li et al. (2018) and Wang & Li (2022), we drop the students with missingness in some important variables, resulting in the treatment group of size $n_1 = 118$ and control group $n_0 = 856$. And we include the following variables to balance: high-school GPA, age, gender and indicators for whether lives at home and whether rarely puts off studying for tests. We exclude GSW due to its incompatibility in dealing with exact imbalanced designs, and include PSRSRR using both $p_a = 10^{-3}$ and $\nu_{p,a} = 0.01$ for threshold selection, while setting $p_a = 10^{-3}$ for other methods.

We generate $10,000$ assignments, and obtain the estimation performance result in Figure 3 and sampling time performance in Table 1. The results show that both selection strategies of PSRSRR have achieved much faster sampling speed than RR and much improved variance reduction compared with CR. Regarding the different strategies used for threshold selection, we can conclude from the results that PSRSRR ($\nu_{p,a}$) is able to reduce most of the variance among all methods, and has an even shorter sampling time compared with PSRSRR ($p_a$). This illustrates the strength of this threshold selection strategy in asymptotic scenarios.

## 5 CONCLUSION AND DISCUSSION

Our study demonstrates that rerandomization, long considered computationally impractical under stringent thresholds, can be made both fast and theoretically sound through a Metropolis–Hastings framework with an importance resampling correction. The key insight is that pair–switching updates naturally form a Markov chain whose stationary distribution favors balanced allocations, and by

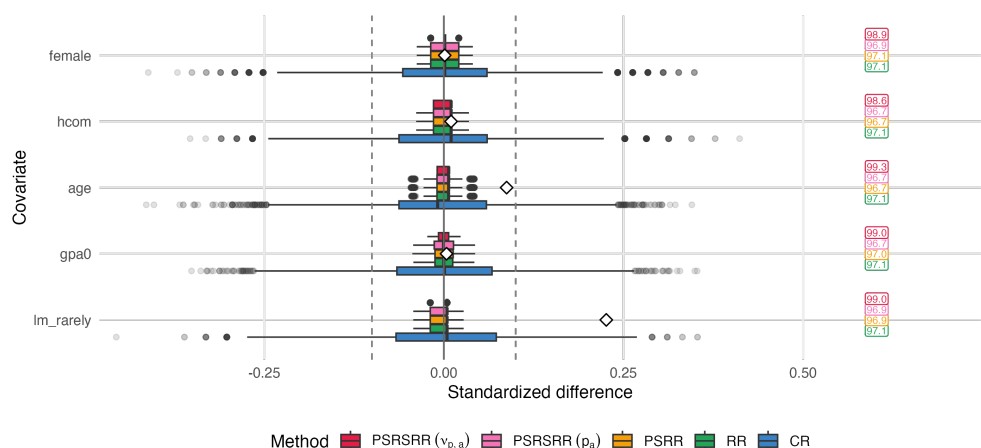

Figure 3: Box-plot of standardized differences in covariate means for STAR data. The diamonds indicates the standardized difference for the actual assignment in the experiment. The values on the right side are PRIVs, the empirical percent reductions in variance compared with CR of each method.

layering rejection sampling we recover exact uniformity over the accepted set. In practice, a simple early–stopping rule yields nearly uniform assignments while accelerating computation by orders of magnitude. Extensive simulations and real–data applications show that this approach preserves the inferential validity of classical rerandomization, narrows confidence intervals, and reduces mean squared error, all while drastically cutting down runtime.

Still, our framework opens several new directions. First, more advanced MCMC kernels may further improve mixing and exploration of the assignment space. Techniques such as adaptive tempering or hybrid proposals (Liang et al., 2010) could be incorporated, potentially achieving better balance or faster convergence. Second, complex experiments are increasingly central to causal inference (Cinelli et al., 2025). Extending our method to factorial, clustered, stratified, or sequential designs will be crucial for ensuring that rerandomization remains feasible and theoretically justified in these contexts. Third, while we improved both the algorithmic efficiency and the implementation via Rcpp, complementary system-level advances are emerging. For example, Goldstein et al. (2025) leverage hardware–accelerated tools for rerandomization and randomization testing. Combining their acceleration with our sampling framework could make strict thresholds practical at scale.

Taken together, our results show that rerandomization need not force a tradeoff between balance and computational feasibility. By unifying fast sampling with rigorous inference guarantees, our approach makes rerandomization a practical tool for modern experimental research, and lays the groundwork for further advances in both methodology and applications.

## REPRODUCIBILITY STATEMENT

The code for reproducing our experiments will be released publicly following the double-blind review process. The technical proofs and details regarding data preprocessing can be found in Appendix A and B, respectively.

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

## A  TECHNICAL PROOFS

### A.1  PROOF OF THEOREM 1

We first give a rigorous mathematical definition of the pair-switching Markov chain constructed in Algorithm 1. For any pair of assignments $\mathbf{W}_i, \mathbf{W}_j \in \mathcal{W}$ and temperature $T$, we use $\mathbb{Q}_T(\mathbf{W}_j|\mathbf{W}_i)$ to denote the transition probability from $\mathbf{W}_i$ to $\mathbf{W}_j$, i.e., the probability that $\mathbf{W}_i$ is updated to $\mathbf{W}_j$ within one step of pair-switching. The update rule indicated by Algorithm 1 could therefore be formulated as follows,

- if $\mathbf{W}_i$ and $\mathbf{W}_j$ are *neighbors*, then

$$\mathbb{Q}_T(\mathbf{W}_j|\mathbf{W}_i) = \frac{1}{n_t n_c} \min\{1, (M(\mathbf{W}_i)/M(\mathbf{W}_j))^{1/T}\}, \tag{2}$$

- if $\mathbf{W}_i$ and $\mathbf{W}_j$ are not neighbors and $\mathbf{W}_i \neq \mathbf{W}_j$, then

$$\mathbb{Q}_T(\mathbf{W}_j|\mathbf{W}_i) = 0, \tag{3}$$

- if $\mathbf{W}_i = \mathbf{W}_j$, then

$$\mathbb{Q}_T(\mathbf{W}_j|\mathbf{W}_i) = 1 - \sum_{k \neq i} \mathbb{Q}_T(\mathbf{W}_k|\mathbf{W}_i), \tag{4}$$

where two assignments are called *neighbors* if and only if one assignment can be obtained by switching one 0-1 pair in the other assignment.

By definition, the sequence $\{\mathbf{W}^{(t)}\}_{t \geq 0}$ constitutes a Markov chain (Givens & Hoeting, 2012, Equation (1.41)). The transition matrix of this chain is given by $\mathbf{Q}_T = (q_{ij})_{m \times m}$, where $q_{ij} = \mathbb{Q}_T(\mathbf{W}_j|\mathbf{W}_i)$ denotes the transition probability from $\mathbf{W}_i$ to $\mathbf{W}_j$. Here, $m = \binom{n}{n_t}$ is the total number of states (i.e., all possible assignments), and $\mathcal{W} = \{\mathbf{W}_1, \mathbf{W}_2, \ldots, \mathbf{W}_m\}$ is the state space.

With the above formulation, we prove Theorem 1 in two steps: (i) verify that the proposed distribution $\pi(\mathbf{W}) \propto M(\mathbf{W})^{-1/T}$ satisfies the detailed balance condition and is therefore the stationary distribution; and (ii) establish the irreducibility and aperiodicity of the Markov chain to guarantee the uniqueness of the stationary distribution and the convergence of the chain to it.

**Step 1.** We explicitly verify that the proposed distribution $\pi$ satisfies the detailed balance condition. Recall from Theorem 1 that the probability mass function is given by:

$$\pi(\mathbf{W}) = \frac{M(\mathbf{W})^{-1/T}}{Z}, \tag{5}$$

where $Z = \sum_{\mathbf{W}^* \in \mathcal{W}} M(\mathbf{W}^*)^{-1/T}$ is the normalizing constant.

We must show that for any pair of states $\mathbf{W}_i, \mathbf{W}_j \in \mathcal{W}$:

$$\pi(\mathbf{W}_i)\mathbb{Q}_T(\mathbf{W}_j|\mathbf{W}_i) = \pi(\mathbf{W}_j)\mathbb{Q}_T(\mathbf{W}_i|\mathbf{W}_j). \tag{6}$$

If $\mathbf{W}_i = \mathbf{W}_j$, the equality holds trivially.

If $\mathbf{W}_i$ and $\mathbf{W}_j$ are not neighbors and $\mathbf{W}_i \neq \mathbf{W}_j$, both sides are zero.

Consider the case where $\mathbf{W}_i$ and $\mathbf{W}_j$ are neighbors. Using the transition probability defined in Equation (2), the left-hand side (LHS) of Equation (6) is:

$$\begin{aligned}
\text{LHS} &= \frac{M(\mathbf{W}_i)^{-1/T}}{Z} \cdot \frac{1}{n_t n_c} \min\left\{1, \left(\frac{M(\mathbf{W}_i)}{M(\mathbf{W}_j)}\right)^{1/T}\right\} \\
&= \frac{1}{Z \cdot n_t n_c} \min\left\{M(\mathbf{W}_i)^{-1/T}, M(\mathbf{W}_i)^{-1/T} \cdot \frac{M(\mathbf{W}_i)^{1/T}}{M(\mathbf{W}_j)^{1/T}}\right\} \\
&= \frac{1}{Z \cdot n_t n_c} \min\left\{M(\mathbf{W}_i)^{-1/T}, M(\mathbf{W}_j)^{-1/T}\right\}.
\end{aligned}$$

By symmetry, the right-hand side (RHS) yields the identical expression:

$$\begin{aligned}
\text{RHS} &= \frac{M(\mathbf{W}_j)^{-1/T}}{Z} \cdot \frac{1}{n_t n_c} \min\left\{1, \left(\frac{M(\mathbf{W}_j)}{M(\mathbf{W}_i)}\right)^{1/T}\right\} \\
&= \frac{1}{Z \cdot n_t n_c} \min\left\{M(\mathbf{W}_j)^{-1/T}, M(\mathbf{W}_i)^{-1/T}\right\}.
\end{aligned}$$

Since LHS = RHS, the detailed balance condition is satisfied, implying that $\pi$ is the stationary distribution (Givens & Hoeting, 2012, Equation (1.43)).

**Step 2.** To establish the *irreducibility*, we denote any two assignments as $\mathbf{W}_i, \mathbf{W}_j \in \mathcal{W}$, and their treatment and control indices are $\mathcal{I}_t, \mathcal{I}_c$ and $\mathcal{J}_t, \mathcal{J}_c$, respectively. The shared indices in the treatment and control groups are denoted as sets $\mathcal{D}_t, \mathcal{D}_c$, respectively. Apparently, we have

$$\mathcal{I}_t \cup \mathcal{I}_c = \mathcal{J}_t \cup \mathcal{J}_c = \{1, 2, \ldots, m\},$$

and further

$$|\mathcal{I}_t \setminus \mathcal{D}_t| = |\mathcal{J}_t \setminus \mathcal{D}_t|, \quad |\mathcal{I}_c \setminus \mathcal{D}_c| = |\mathcal{J}_c \setminus \mathcal{D}_c|.$$

For the treatment indices of $\mathbf{W}_i$, if they are not shared by treatment indices of $\mathbf{W}_j$, then they will be included in the control indices of $\mathbf{W}_j$ not shared by control indices of $\mathbf{W}_i$, since treatment and control groups have no overlap, i.e., $\mathcal{I}_t \setminus \mathcal{D}_t \subseteq \mathcal{J}_c \setminus \mathcal{D}_c$. We can similarly obtain the conclusion that $\mathcal{I}_c \setminus \mathcal{D}_c \subseteq \mathcal{J}_t \setminus \mathcal{D}_t$. Therefore, we have

$$|\mathcal{I}_t \setminus \mathcal{D}_t| = |\mathcal{J}_t \setminus \mathcal{D}_t| = |\mathcal{I}_c \setminus \mathcal{D}_c| = |\mathcal{J}_c \setminus \mathcal{D}_c|,$$

indicating that pairs could be formed between $\mathcal{I}_t \setminus \mathcal{D}_t$ and $\mathcal{I}_c \setminus \mathcal{D}_c$. So after switching each pair, whose probability is positive as shown in the formulation of the transition matrix, the generated $\mathbf{W}_{i'}$ would be equal to $\mathbf{W}_j$, which verifies that any two states in this Markov chain could communicate with each other.

To establish the *aperiodicity*, we need to show that $\mathbf{Q}_T^2$ and $\mathbf{Q}_T^3$ both have positive diagonal elements, implying that the period is given by $\gcd(2,3) = 1$, where $\gcd()$ denotes the greatest common divisor function. The property of $\mathbf{Q}_T^2$ could be readily verified: as for any $\mathbf{W}_i \in \mathcal{W}$, the assignment can return to itself by switching the 0-1 pair to become its neighbor and then reversing that switch, with transition probabilities both greater than zero. As for $\mathbf{Q}_T^3$, for any $\mathbf{W}_i \in \mathcal{W}$, denote $p, s$ as the indices in the control group and $q$ the index in the treatment group. Since transition probability between any neighbor is positive, we can easily verify the conclusion by the following derivation. First,

the switch takes place between 0-1 pair $(p, q)$. Then the second switch is performed between $(p, s)$, and the third between $(q, s)$, returning to the initial $\mathbf{W}_i$. Therefore, we confirm the aperiodicity of the chain.

Having established the irreducibility and aperiodicity of the Markov chain $\{\mathbf{W}^{(t)}\}_{t \geq 0}$, we invoke Lemma 4 to conclude that the stationary distribution $\pi$ derived in Step 1 is unique and equal to the limiting distribution. This completes the proof of Theorem 1.

**Lemma 4** *(Summary theorem for ergodic, finite-state discrete-time Markov chains, Theorem 25.19 in Harchol-Balter (2023)) Given an aperiodic and irreducible finite-state chain, the following results hold:*

- *The limiting distribution exists and has all-positive components.*

- *The stationary distribution is unique and is equal to the limiting distribution.*

### A.2 PROOF OF THEOREM 2

We follow the steps given in Section 6.2.3 in Givens & Hoeting (2012) to give this proof.

Our target distribution is a uniform distribution on $\mathcal{W}_a$, i.e., the probability mass function would be $f(\mathbf{W}) = \frac{\mathbb{I}\{\mathbf{W} \in \mathcal{W}_a\}}{|\mathcal{W}_a|}$, and $\pi(\mathbf{W})$ denote the stationary distribution sampled from Algorithm 1. Let $e(\mathbf{W})$ denote an envelope function, and in this case, we have

$$e(\mathbf{W}) = \frac{\pi(\mathbf{W})}{\alpha} = \frac{M(\mathbf{W})^{-1/T}}{\alpha \cdot \sum_{\mathbf{W}^* \in \mathcal{W}} M(\mathbf{W}^*)^{-1/T}} \geq f(\mathbf{W}) = \frac{\mathbb{I}\{\mathbf{W} \in \mathcal{W}_a\}}{|\mathcal{W}_a|}$$

where $\alpha$ is a scaling parameter, and we choose

$$\alpha = \frac{a^{-1/T} |\mathcal{W}_a|}{\sum_{\mathbf{W}^* \in \mathcal{W}} M(\mathbf{W}^*)^{-1/T}}.$$

We go through the following sampling procedure.

1. Sample $\mathbf{Y} \sim \pi$ which is the stationary distribution;

2. Sample $J \sim \mathcal{B}er\left(\frac{f(\mathbf{Y})}{e(\mathbf{Y})}\right)$, where

$$\begin{aligned}
\frac{f(\mathbf{Y})}{e(\mathbf{Y})} &= \frac{\mathbb{I}\{\mathbf{Y} \in \mathcal{W}_a\}}{|\mathcal{W}_a|} \frac{\sum_{\mathbf{W}^* \in \mathcal{W}} M(\mathbf{W}^*)^{-1/T}}{M(\mathbf{Y})^{-1/T}} \frac{a^{-1/T} |\mathcal{W}_a|}{\sum_{\mathbf{W}^* \in \mathcal{W}} M(\mathbf{W}^*)^{-1/T}} \\
&= \frac{M(\mathbf{Y})^{1/T} \cdot \mathbb{I}\{\mathbf{Y} \in \mathcal{W}_a\}}{a^{1/T}} \\
&= \mathbb{I}\{\mathbf{Y} \in \mathcal{W}_a\} \left(\frac{M(\mathbf{Y})}{a}\right)^{1/T} \\
&= p(\mathbf{Y})
\end{aligned}$$

which is the acceptance probability as defined in Algorithm 2;

3. Reject $\mathbf{Y}$ if $J = 0$, and do not record $\mathbf{Y}$ but instead return to step 1; Otherwise, keep the value of $\mathbf{Y}$, set $\mathbf{W} = \mathbf{Y}$.

We verify that the above sampling procedure could indeed generate the targeted distribution,

$$
\begin{aligned}
\mathbb{P}(\mathbf{W} = y) &= \mathbb{P}(\mathbf{Y} = y \mid J = 1) \\
&= \frac{\mathbb{P}(\mathbf{Y} = y, J = 1)}{\mathbb{P}(J = 1)} \\
&= \frac{\pi(y) \cdot \frac{f(y)}{e(y)}}{\sum_{z \in \mathcal{W}} \pi(z) \cdot \frac{f(z)}{e(z)}} \\
&= \frac{\pi(y) \cdot \alpha \cdot \frac{f(y)}{\pi(y)}}{\sum_{z \in \mathcal{W}} \pi(z) \cdot \alpha \cdot \frac{f(z)}{\pi(z)}} \\
&= \frac{\alpha f(y)}{\alpha \sum_{z \in \mathcal{W}} f(z)} \\
&= f(y).
\end{aligned}
$$

Therefore, we conclude the proof. $\qquad\square$

### A.3 THEORETICAL ANALYSIS OF THE GENERALIZED FRAMEWORK

In Algorithm 3, at each step $t$, we check if the current state $\mathbf{W}^{(t)}$ can be accepted via rejection sampling. We can model this check as a probabilistic filter. Let the acceptance probability function $p_{\text{acc}} : \mathcal{W} \to [0, 1]$ be defined as:

$$
p_{\text{acc}}(\mathbf{W}) = \mathbb{I}\{\mathbf{W} \in \mathcal{W}_a\} \left( \frac{M(\mathbf{W})}{a} \right)^{1/T}, \quad \text{and} \quad p_{\text{rej}}(\mathbf{W}) = 1 - p_{\text{acc}}(\mathbf{W}).
$$

Throughout this analysis, we use the term "check" to denote the procedure of attempting to accept $\mathbf{W}$ with probability $p_{\text{acc}}(\mathbf{W})$ (upon which the algorithm terminates) or rejecting it with probability $p_{\text{rej}}(\mathbf{W})$ (upon which the chain continues).

We will analyze a generalized version of the algorithm characterized by two parameters: an explicit "burn-in" period of length $L_{\text{burn}}$, and a check interval $s_{\text{chk}}$. In this generalized process, the chain evolves for $L_{\text{burn}}$ steps without any checks (i.e., without attempting rejection sampling), and subsequently conducts a check only every $s_{\text{chk}}$ steps. Intuitively, a sufficiently large $L_{\text{burn}}$ ensures the chain initially converges to stationarity, while a large $s_{\text{chk}}$ allows the chain to recover from the distributional distortion caused by rejected checks. We will establish conditions under which this process contracts toward the target distribution despite the distributional distortion introduced by these periodic rejection sampling checks.

In this theoretical analysis, we assume the Markov chain evolves according to a "lazy" version of the Metropolis-Hastings kernel, denoted $\mathbf{Q}_{\text{lazy}} = \frac{1}{2}\mathbf{I} + \frac{1}{2}\mathbf{Q}_T$, rather than the kernel $\mathbf{Q}_T$ used by Algorithm 3. Probabilistically, this corresponds to a process that stays in the current state with probability $1/2$ and transitions according to $\mathbf{Q}_T$ with probability $1/2$. This modification ensures that all eigenvalues of the transition matrix are non-negative, which makes our analysis convenient. Let $1 = \lambda_1 > \lambda_2 \geq \cdots \geq \lambda_m \geq 0$ denote these eigenvalues. We define the spectral gap as $\gamma_{\text{MH}} = 1 - \lambda_2$, which governs the geometric rate of convergence to stationarity. Since the actual non-lazy algorithm moves more frequently, it mixes faster than this theoretical lazy version. Therefore, the convergence guarantees derived for the lazy kernel naturally hold for the non-lazy one.

Let $\mu$ be a probability distribution on $\mathcal{W}$, where $\mu(\mathbf{W})$ denotes the probability mass of state $\mathbf{W}$. When a check is performed, the distribution splits into two components based on the acceptance probability $p_{\text{acc}}(\mathbf{W})$ and rejection probability $p_{\text{rej}}(\mathbf{W}) = 1 - p_{\text{acc}}(\mathbf{W})$. We formalize the resulting conditional distributions as reweighting operators.

**Definition A.1 (Reweighting Operators)** *For a probability distribution $\mu$, we define the Acceptance Operator $\mathcal{T}_{\text{acc}}$ and the Rejection Operator $\mathcal{T}_{\text{rej}}$ pointwise as:*

$$
\mathcal{T}_{\text{acc}}(\mu)(\mathbf{W}) = \frac{\mu(\mathbf{W}) p_{\text{acc}}(\mathbf{W})}{\mathbb{E}_\mu[p_{\text{acc}}]}, \quad \mathcal{T}_{\text{rej}}(\mu)(\mathbf{W}) = \frac{\mu(\mathbf{W}) p_{\text{rej}}(\mathbf{W})}{\mathbb{E}_\mu[p_{\text{rej}}]},
$$

*where the denominators $\mathbb{E}_\mu[p_{\mathrm{acc}}]$ and $\mathbb{E}_\mu[p_{\mathrm{rej}}]$ are the normalizing constants representing the total probability of acceptance and rejection, respectively.*

We define the Block Map $\Phi_{\mathrm{blk}}$ to describe the evolution of the chain's distribution between consecutive checks. Consider a cycle starting with a distribution $\mu$. First, the chain evolves for $s_{\mathrm{chk}}$ steps according to the lazy kernel $\mathbf{Q}_{\mathrm{lazy}}$, transforming the distribution to $\mu\mathbf{Q}_{\mathrm{lazy}}^{s_{\mathrm{chk}}}$. Then, a check is performed. If the state is rejected (which is necessary for the chain to continue), the distribution is updated by the rejection operator $\mathcal{T}_{\mathrm{rej}}$. The composition of these steps yields the new distribution:

$$\Phi_{\mathrm{blk}}(\mu) = \mathcal{T}_{\mathrm{rej}}\left(\mu\mathbf{Q}_{\mathrm{lazy}}^{s_{\mathrm{chk}}}\right).$$

Intuitively, a sufficiently large $L_{\mathrm{burn}}$ ensures the chain initially converges to stationarity, while a sufficiently large $s_{\mathrm{chk}}$ ensures that the mixing within each block is strong enough to counteract the distributional distortion caused by the rejection step. The following Theorem 5 formalizes this intuition, providing a deterministic bound on the total variation (TV) distance between the output and the target uniform distribution.

To state the result, we first define the necessary parameters. Let $p_{\mathrm{acc}}^\star = \mathbb{E}_{\mathbf{W}\sim\pi}[p_{\mathrm{acc}}(\mathbf{W})]$ be the stationary acceptance probability. Fix a target upper bound $\epsilon \in (0,1)$ for the total variation distance between the output and the target uniform distribution. To achieve this target, the chain must satisfy a tighter proximity condition before the check, which we denote by $\delta_\epsilon = \frac{\epsilon p_{\mathrm{acc}}^\star}{2(1+\epsilon)}$.

Our analysis tracks the $\chi^2$-divergence from stationarity, as it provides a tractable upper bound on the total variation distance and allows us to quantify the interplay between mixing and rejection. The rejection sampling step tends to push the distribution away from stationarity (distortion), while the mixing steps pull it back (contraction). For a check interval $s$ and precision $\delta$, we define the distortion coefficients $A_{\mathrm{blk}}$ and $B_{\mathrm{rej}}$:

$$A_{\mathrm{blk}}(s,\delta) = \frac{(1-\gamma_{\mathrm{MH}})^{2s}}{(1-p_{\mathrm{acc}}^\star - 2\delta)^2}, \quad B_{\mathrm{rej}}(\delta) = \frac{1}{(1-p_{\mathrm{acc}}^\star - 2\delta)^2} - 1.$$

The stable radius $R_{\mathrm{blk}}$ represents the fixed point of this process—if the chain enters this $\chi^2$-neighborhood of stationarity, the mix-then-reject cycle ensures it remains there:

$$R_{\mathrm{blk}}(s,\delta) = \sqrt{\frac{B_{\mathrm{rej}}(\delta)}{1-A_{\mathrm{blk}}(s,\delta)}}.$$

**Theorem 5** *Suppose the burn-in length $L_{\mathrm{burn}}$ and check interval $s_{\mathrm{chk}}$ satisfy the following three conditions:*

*(i) Entry Condition: The burn-in $L_{\mathrm{burn}}$ is sufficiently long to bring the initial distribution $\mathrm{Unif}(\mathcal{W})$ close to stationarity. Specifically:*

$$L_{\mathrm{burn}} \geq \frac{1}{\gamma_{\mathrm{MH}}}\log\left(\frac{\rho_{\chi^2}^{init}}{R_{\mathrm{blk}}(s_{\mathrm{chk}},\delta_\epsilon)}\right),$$

*where $\rho_{\chi^2}^{init}$ is the initial $\chi^2$-distance from stationarity.*

*(ii) Block Contraction: The mixing over $s_{\mathrm{chk}}$ steps dominates the rejection distortion, ensuring the process remains stable:*

$$(1-\gamma_{\mathrm{MH}})^{s_{\mathrm{chk}}} < 1 - p_{\mathrm{acc}}^\star - 2\delta_\epsilon.$$

*(iii) Pre-check Contraction: The mixing within a block ensures the distribution is sufficiently close to stationarity immediately before a check is performed:*

$$(1-\gamma_{\mathrm{MH}})^{s_{\mathrm{chk}}} R_{\mathrm{blk}}(s_{\mathrm{chk}},\delta_\epsilon) \leq 2\delta_\epsilon.$$

*Then the distribution of the output $\mathbf{W}_{out}$ satisfies:*

$$d_{TV}(\mathrm{Law}(\mathbf{W}_{out}), \mathrm{Unif}(\mathcal{W}_a)) \leq \epsilon.$$

The proof will track the distance to stationarity using the $\chi^2$-divergence. We will use several technical lemmas, whose proofs are all relegated to Appendix A.4.

We define $\mu_t$ as the distribution of the chain at step $t$ conditional on not having accepted prior to $t$. We analyze the process at the check times $t_k = L_{\text{burn}} + k \cdot s_{\text{chk}}$.

First, we must ensure the chain is close to stationarity before the first check is performed. We rely on the following mixing bound for the lazy kernel.

**Lemma 6** *For any distribution $\mu$ and any number of steps $L \geq 1$, the lazy Metropolis-Hastings kernel $\mathbf{Q}_{\text{lazy}}$ contracts the $\chi^2$-divergence toward stationarity:*

$$\chi^2(\mu \mathbf{Q}_{\text{lazy}}^L \parallel \pi) \leq (1 - \gamma_{\text{MH}})^{2L} \cdot \chi^2(\mu \parallel \pi).$$

Let $\rho_{\chi^2}^{\text{init}}$ denote the initial distance. Applying Lemma 6, the distance after $L_{\text{burn}}$ steps is $\chi^2(\mu_{L_{\text{burn}}} \parallel \pi) \leq (1 - \gamma_{\text{MH}})^{2L_{\text{burn}}}(\rho_{\chi^2}^{\text{init}})^2$. The Entry Condition (ii) implies that $(1 - \gamma_{\text{MH}})^{L_{\text{burn}}} \rho_{\chi^2}^{\text{init}} \leq e^{-\gamma_{\text{MH}} \cdot L_{\text{burn}}} \rho_{\chi^2}^{\text{init}} \leq R_{\text{blk}}$. Therefore, the chain enters the stable region at the first check:

$$\chi^2(\mu_{L_{\text{burn}}} \parallel \pi) \leq R_{\text{blk}}^2(s_{\text{chk}}, \delta_\epsilon). \tag{7}$$

Next, we ensure the chain stays close to stationarity despite the distortion introduced by rejected checks. The evolution of the chain between failed checks corresponds to the Block Map $\Phi_{\text{blk}}(\mu) = \mathcal{T}_{\text{rej}}(\mu \mathbf{Q}_{\text{lazy}}^{s_{\text{chk}}})$. We use the following lemma.

**Lemma 7** *If the number of steps $s_{\text{chk}}$ is large enough such that Conditions (ii) and (iii) hold, then the map $\Phi_{\text{blk}}$ maps the $\chi^2$-ball of radius $R_{\text{blk}}$ into itself. Specifically, if $\chi^2(\mu \parallel \pi) \leq R_{\text{blk}}^2$, then:*

$$\chi^2(\Phi_{\text{blk}}(\mu) \parallel \pi) \leq R_{\text{blk}}^2.$$

By induction, starting from equation 7 and applying Lemma 7, we guarantee that for all check times $t_k \geq L_{\text{burn}}$, the distribution remains bounded:

$$\chi^2(\mu_{t_k} \parallel \pi) \leq R_{\text{blk}}^2. \tag{8}$$

Finally, we analyze the final block where acceptance occurs. Suppose the algorithm accepts at time $t^* = t_k + s_{\text{chk}}$. Let $\mu_{t_k}$ be the distribution at the start of this final block. We have shown that $\chi^2(\mu_{t_k} \parallel \pi) \leq R_{\text{blk}}^2$. The distribution *immediately before* the acceptance check is $\nu = \mu_{t_k} Q_{\text{lazy}}^{s_{\text{chk}}}$. By Lemma 6, the mixing step contracts the distance:

$$\chi^2(\nu \parallel \pi) \leq (1 - \gamma_{\text{MH}})^{2s_{\text{chk}}} \chi^2(\mu_{t_k} \parallel \pi) \leq (1 - \gamma_{\text{MH}})^{2s_{\text{chk}}} R_{\text{blk}}^2.$$

We convert this to Total Variation distance to verify the condition for the Acceptance Lemma:

$$d_{TV}(\nu, \pi) \leq \frac{1}{2}\sqrt{\chi^2(\nu \parallel \pi)} \leq \frac{1}{2}(1 - \gamma_{\text{MH}})^{s_{\text{chk}}} R_{\text{blk}}.$$

Condition (iii) ensures that this quantity is bounded by $\delta_\epsilon$. Thus, $d_{TV}(\nu, \pi) \leq \delta_\epsilon$. We can now apply Lemma 8 to the pre-check distribution $\nu$.

**Lemma 8** *If the pre-check distribution $\mu$ satisfies $d_{TV}(\mu, \pi) \leq \delta$, then the output distribution satisfies:*

$$d_{TV}(\mathcal{T}_{\text{acc}}(\mu), \text{Unif}(\mathcal{W}_a)) \leq \frac{2\delta}{p_{\text{acc}}^\star - 2\delta}.$$

Substituting $\delta_\epsilon = \frac{\epsilon p_{\text{acc}}^\star}{2(1+\epsilon)}$ into Lemma 8, we obtain:

$$d_{TV}(\text{Law}(\mathbf{W}_{\text{out}}), \text{Unif}(\mathcal{W}_a)) \leq \frac{2\left(\frac{\epsilon p_{\text{acc}}^\star}{2(1+\epsilon)}\right)}{p_{\text{acc}}^\star - 2\left(\frac{\epsilon p_{\text{acc}}^\star}{2(1+\epsilon)}\right)} = \epsilon.$$

This completes the proof of the theorem.

### A.4 PROOFS OF TECHNICAL LEMMAS IN A.3

#### A.4.1 PROOF OF LEMMA 6

By definition, the $\chi^2$-divergence is the expected squared deviation of the density ratio under the stationary distribution:

$$\chi^2(\mu \parallel \pi) = \mathbb{E}_{\mathbf{W} \sim \pi}\left[\left(\frac{d\mu}{d\pi}(\mathbf{W}) - 1\right)^2\right].$$

The transition kernel $\mathbf{Q}_{\text{lazy}}$ acts as a linear operator on functions of the state space. If the initial density relative to $\pi$ is $\frac{d\mu}{d\pi}$, then after $L$ steps, the new density is $\mathbf{Q}_{\text{lazy}}^L(\frac{d\mu}{d\pi})$. Since the kernel preserves constants ($\mathbf{Q}_{\text{lazy}}\mathbf{1} = \mathbf{1}$), the deviation from 1 evolves linearly:

$$\frac{d(\mu\mathbf{Q}_{\text{lazy}}^L)}{d\pi} - 1 = \mathbf{Q}_{\text{lazy}}^L\left(\frac{d\mu}{d\pi}\right) - 1 = \mathbf{Q}_{\text{lazy}}^L\left(\frac{d\mu}{d\pi} - 1\right).$$

Consequently, the $\chi^2$-divergence after $L$ steps is the second moment of this propagated deviation:

$$\chi^2(\mu\mathbf{Q}_{\text{lazy}}^L \parallel \pi) = \mathbb{E}_\pi\left[\left(\mathbf{Q}_{\text{lazy}}^L\left(\frac{d\mu}{d\pi} - 1\right)\right)^2\right].$$

We now bound this expectation. The function $\left(\frac{d\mu}{d\pi} - 1\right)$ has mean zero under $\pi$ because $\mathbb{E}_\pi[\frac{d\mu}{d\pi}] = 1$. Therefore, it is orthogonal to the constant eigenfunction corresponding to the largest eigenvalue $\lambda_1 = 1$.

Since the kernel is lazy, all eigenvalues are non-negative, and the operator $\mathbf{Q}_{\text{lazy}}$ contracts the magnitude of any mean-zero function by at least the spectral gap factor $\lambda_2 = 1 - \gamma_{\text{MH}}$ at each step. Thus:

$$\mathbb{E}_\pi\left[\left(\mathbf{Q}_{\text{lazy}}^L\left(\frac{d\mu}{d\pi} - 1\right)\right)^2\right] \leq (1 - \gamma_{\text{MH}})^{2L}\,\mathbb{E}_\pi\left[\left(\frac{d\mu}{d\pi} - 1\right)^2\right].$$

Substituting the definition of $\chi^2$-divergence back into the RHS completes the proof:

$$\chi^2(\mu\mathbf{Q}_{\text{lazy}}^L \parallel \pi) \leq (1 - \gamma_{\text{MH}})^{2L}\chi^2(\mu \parallel \pi).$$

#### A.4.2 PROOF OF LEMMA 7

The Block Map is the composition $\Phi_{\text{blk}}(\mu) = \mathcal{T}_{\text{rej}}(\nu)$, where $\nu = \mu\mathbf{Q}_{\text{lazy}}^{s_{\text{chk}}}$. We analyze the evolution of the $\chi^2$-divergence in two steps.

Step 1: Mixing Phase. Assume the initial distribution satisfies $\chi^2(\mu \parallel \pi) \leq R_{\text{blk}}^2$. By Lemma 6, applying the lazy kernel for $s_{\text{chk}}$ steps contracts the distance:

$$\chi^2(\nu \parallel \pi) \leq (1 - \gamma_{\text{MH}})^{2s_{\text{chk}}}\chi^2(\mu \parallel \pi) \leq (1 - \gamma_{\text{MH}})^{2s_{\text{chk}}}R_{\text{blk}}^2. \tag{9}$$

Using the bound $d_{TV}(\nu, \pi) \leq \frac{1}{2}\sqrt{\chi^2(\nu \parallel \pi)}$, we have:

$$d_{TV}(\nu, \pi) \leq \frac{1}{2}(1 - \gamma_{\text{MH}})^{s_{\text{chk}}}R_{\text{blk}}.$$

Condition (ii) explicitly requires that $(1 - \gamma_{\text{MH}})^{s_{\text{chk}}}R_{\text{blk}} \leq 2\delta$. Therefore, $d_{TV}(\nu, \pi) \leq \delta$.

Step 2: Rejection Phase. We will use the following lemma, which quantifies the "distortion" introduced by the rejection step.

**Lemma 9** *If a distribution $\mu$ satisfies $d_{TV}(\mu, \pi) \leq \delta$ for some $\delta < (1 - p_{\text{acc}}^\star)/2$, then applying the rejection operator increases the $\chi^2$-divergence from stationarity by at most:*

$$\chi^2(\mathcal{T}_{\text{rej}}(\mu) \parallel \pi) \leq A_{\text{rej}}(\delta) \cdot \chi^2(\mu \parallel \pi) + B_{\text{rej}}(\delta),$$

*where $A_{\text{rej}}(\delta) = B_{\text{rej}}(\delta) + 1 = \frac{1}{(1 - p_{\text{acc}}^\star - 2\delta)^2}$.*

Substituting the bound from Equation 9 into Lemma 9:

$$\chi^2(\Phi_{\text{blk}}(\mu) \parallel \pi) \leq A_{\text{rej}}(\delta) \left[ (1 - \gamma_{\text{MH}})^{2s_{\text{chk}}} R_{\text{blk}}^2 \right] + B_{\text{rej}}(\delta).$$

We recognize the coefficient of $R_{\text{blk}}^2$ as the definition of $A_{\text{blk}}(s_{\text{chk}}, \delta)$ from Theorem 5:

$$A_{\text{blk}}(s_{\text{chk}}, \delta) = A_{\text{rej}}(\delta)(1 - \gamma_{\text{MH}})^{2s_{\text{chk}}}.$$

Thus, the recurrence relation is:

$$\chi^2(\Phi_{\text{blk}}(\mu) \parallel \pi) \leq A_{\text{blk}}(s_{\text{chk}}, \delta) R_{\text{blk}}^2 + B_{\text{rej}}(\delta).$$

**Step 3: Fixed Point Verification.** To prove the lemma, we must show that the RHS is at most $R_{\text{blk}}^2$. We substitute the definition of the stable radius $R_{\text{blk}} = \sqrt{\frac{B_{\text{rej}}(\delta)}{1 - A_{\text{blk}}(s_{\text{chk}}, \delta)}}$:

$$A_{\text{blk}} \left( \frac{B_{\text{rej}}}{1 - A_{\text{blk}}} \right) + B_{\text{rej}} = B_{\text{rej}} \left( \frac{A_{\text{blk}}}{1 - A_{\text{blk}}} + 1 \right) = B_{\text{rej}} \left( \frac{A_{\text{blk}} + (1 - A_{\text{blk}})}{1 - A_{\text{blk}}} \right) = \frac{B_{\text{rej}}}{1 - A_{\text{blk}}} = R_{\text{blk}}^2.$$

Therefore, $\chi^2(\Phi_{\text{blk}}(\mu) \parallel \pi) \leq R_{\text{blk}}^2$. The map contracts the ball into itself.

### A.4.3 PROOF OF LEMMA 8

From Theorem 2, we know that the target uniform distribution is exactly the result of applying the acceptance operator to the stationary distribution $\pi$. That is:

$$\text{Unif}(\mathcal{W}_a) = \mathcal{T}_{\text{acc}}(\pi).$$

Therefore, our goal is to bound the total variation distance $d_{TV}(\mathcal{T}_{\text{acc}}(\mu), \mathcal{T}_{\text{acc}}(\pi))$.

Let $Z_\mu = \mathbb{E}_{\mathbf{W} \sim \mu}[p_{\text{acc}}(\mathbf{W})]$ and $Z_\pi = \mathbb{E}_{\mathbf{W} \sim \pi}[p_{\text{acc}}(\mathbf{W})] = p_{\text{acc}}^\star$ be the normalization constants (acceptance probabilities) for the respective distributions.

First, we bound the difference between these constants. By the definition of total variation distance $d_{TV}(\mu, \pi) = \frac{1}{2} \sum_{\mathbf{W}} |\mu(\mathbf{W}) - \pi(\mathbf{W})|$, and noting that $0 \leq p_{\text{acc}}(\mathbf{W}) \leq 1$, we have:

$$|Z_\mu - Z_\pi| = \left| \sum_{\mathbf{W} \in \mathcal{W}} p_{\text{acc}}(\mathbf{W})(\mu(\mathbf{W}) - \pi(\mathbf{W})) \right| \leq \sum_{\mathbf{W} \in \mathcal{W}} 1 \cdot |\mu(\mathbf{W}) - \pi(\mathbf{W})| = 2d_{TV}(\mu, \pi) \leq 2\delta.$$

This implies a lower bound on the perturbed acceptance rate:

$$Z_\mu \geq Z_\pi - |Z_\mu - Z_\pi| \geq p_{\text{acc}}^\star - 2\delta.$$

Next, we analyze the total variation distance between the reweighted distributions. The probability mass function of $\mathcal{T}_{\text{acc}}(\mu)$ is $\frac{p_{\text{acc}}(\mathbf{W})\mu(\mathbf{W})}{Z_\mu}$. We decompose the difference:

$$\begin{aligned}
d_{TV}(\mathcal{T}_{\text{acc}}(\mu), \mathcal{T}_{\text{acc}}(\pi)) &= \frac{1}{2} \sum_{\mathbf{W} \in \mathcal{W}} \left| \frac{p_{\text{acc}}(\mathbf{W})\mu(\mathbf{W})}{Z_\mu} - \frac{p_{\text{acc}}(\mathbf{W})\pi(\mathbf{W})}{Z_\pi} \right| \\
&= \frac{1}{2} \sum_{\mathbf{W} \in \mathcal{W}} p_{\text{acc}}(\mathbf{W}) \left| \frac{\mu(\mathbf{W})}{Z_\mu} - \frac{\pi(\mathbf{W})}{Z_\pi} \right| \\
&= \frac{1}{2} \sum_{\mathbf{W} \in \mathcal{W}} p_{\text{acc}}(\mathbf{W}) \left| \frac{\mu(\mathbf{W})}{Z_\mu} - \frac{\pi(\mathbf{W})}{Z_\mu} + \frac{\pi(\mathbf{W})}{Z_\mu} - \frac{\pi(\mathbf{W})}{Z_\pi} \right|.
\end{aligned}$$

Using the triangle inequality and the fact that $p_{\text{acc}}(\mathbf{W}) \leq 1$:

$$\begin{aligned}
d_{TV}(\dots) &\leq \frac{1}{2} \left( \sum_{\mathbf{W}} \frac{p_{\text{acc}}(\mathbf{W})}{Z_\mu} |\mu(\mathbf{W}) - \pi(\mathbf{W})| + \sum_{\mathbf{W}} p_{\text{acc}}(\mathbf{W})\pi(\mathbf{W}) \left| \frac{1}{Z_\mu} - \frac{1}{Z_\pi} \right| \right) \\
&\leq \frac{1}{2Z_\mu} \underbrace{\sum_{\mathbf{W}} |\mu(\mathbf{W}) - \pi(\mathbf{W})|}_{2d_{TV}(\mu, \pi)} + \frac{1}{2} \left| \frac{1}{Z_\mu} - \frac{1}{Z_\pi} \right| \underbrace{\sum_{\mathbf{W}} p_{\text{acc}}(\mathbf{W})\pi(\mathbf{W})}_{Z_\pi}.
\end{aligned}$$

Simplifying the second term:

$$\frac{1}{2}Z_\pi \left| \frac{Z_\pi - Z_\mu}{Z_\mu Z_\pi} \right| = \frac{1}{2Z_\mu}|Z_\pi - Z_\mu|.$$

Substituting the bounds we derived earlier ($d_{TV} \leq \delta$ and $|Z_\pi - Z_\mu| \leq 2\delta$):

$$d_{TV}(\mathcal{T}_{\text{acc}}(\mu), \text{Unif}(\mathcal{W}_a)) \leq \frac{1}{2Z_\mu}(2\delta) + \frac{1}{2Z_\mu}(2\delta) = \frac{2\delta}{Z_\mu}.$$

Finally, using the lower bound $Z_\mu \geq p^\star_{\text{acc}} - 2\delta$, we obtain:

$$d_{TV}(\mathcal{T}_{\text{acc}}(\mu), \text{Unif}(\mathcal{W}_a)) \leq \frac{2\delta}{p^\star_{\text{acc}} - 2\delta}.$$

### A.4.4    PROOF OF LEMMA 9

Let $g(\mathbf{W}) = \frac{d\mu}{d\pi}(\mathbf{W})$ be the density ratio of $\mu$ with respect to $\pi$. The $\chi^2$-divergence is given by:

$$\chi^2(\mu \parallel \pi) = \mathbb{E}_\pi[g(\mathbf{W})^2] - 1.$$

The Rejection Operator $\mathcal{T}_{\text{rej}}$ reweights the distribution by $p_{\text{rej}}(\mathbf{W}) = 1 - p_{\text{acc}}(\mathbf{W})$. The Radon-Nikodym derivative of the new distribution $\mu' = \mathcal{T}_{\text{rej}}(\mu)$ with respect to $\pi$ is:

$$\frac{d\mu'}{d\pi}(\mathbf{W}) = \frac{p_{\text{rej}}(\mathbf{W})g(\mathbf{W})}{Z_\mu},$$

where $Z_\mu = \mathbb{E}_\mu[p_{\text{rej}}(\mathbf{W})]$ is the probability of rejection under $\mu$.

We first derive the expression for the new $\chi^2$-divergence:

$$\chi^2(\mu' \parallel \pi) = \mathbb{E}_\pi\left[\left(\frac{p_{\text{rej}}(\mathbf{W})g(\mathbf{W})}{Z_\mu}\right)^2\right] - 1$$

$$= \frac{1}{Z_\mu^2}\mathbb{E}_\pi\left[p_{\text{rej}}(\mathbf{W})^2 g(\mathbf{W})^2\right] - 1.$$

Since $0 \leq p_{\text{rej}}(\mathbf{W}) \leq 1$, we have $p_{\text{rej}}(\mathbf{W})^2 \leq 1$. Therefore:

$$\mathbb{E}_\pi\left[p_{\text{rej}}(\mathbf{W})^2 g(\mathbf{W})^2\right] \leq \mathbb{E}_\pi\left[g(\mathbf{W})^2\right] = \chi^2(\mu \parallel \pi) + 1.$$

Substituting this back into the divergence equation:

$$\chi^2(\mu' \parallel \pi) \leq \frac{1}{Z_\mu^2}\left(\chi^2(\mu \parallel \pi) + 1\right) - 1 = \frac{1}{Z_\mu^2}\chi^2(\mu \parallel \pi) + \left(\frac{1}{Z_\mu^2} - 1\right).$$

To complete the proof, we must lower-bound the rejection probability $Z_\mu$. Let $Z_\pi = \mathbb{E}_\pi[p_{\text{rej}}(\mathbf{W})] = p^\star_{\text{rej}}$. Using the definition of total variation distance:

$$|Z_\mu - Z_\pi| = |\mathbb{E}_\mu[p_{\text{rej}}] - \mathbb{E}_\pi[p_{\text{rej}}]| \leq 2d_{TV}(\mu, \pi).$$

By assumption, $d_{TV}(\mu, \pi) \leq \delta$. Thus:

$$Z_\mu \geq Z_\pi - 2\delta = p^\star_{\text{rej}} - 2\delta.$$

Provided $\delta < p^\star_{\text{rej}}/2$, the denominator is positive. Setting $A_{\text{rej}}(\delta) = \frac{1}{(p^\star_{\text{rej}} - 2\delta)^2}$ and $B_{\text{rej}}(\delta) = A_{\text{rej}}(\delta) - 1$, we obtain:

$$\chi^2(\mathcal{T}_{\text{rej}}(\mu) \parallel \pi) \leq A_{\text{rej}}(\delta)\chi^2(\mu \parallel \pi) + B_{\text{rej}}(\delta).$$

# B ADDITIONAL EXPERIMENT DETAILS AND RESULTS

## B.1 DETAILED SIMULATION SETTINGS

When conducting simulation studies to examine the performance of estimation results as well as sampling speeds, we use different sample sizes $n$ and their corresponding sets of number of covariates $p$ as shown in Table 2 below. The range of sample size $n$ could help illustrate the performance of our proposed method in a larger scale, from small-sample behavior to large-sample behavior. And for each sample size $n$, we choose different values of $p$, while not violating the asymptotic rule, $p = O(\log n)$, as demonstrated in Harshaw et al. (2024).

Table 2: Simulation setting regarding sample size and number of covariates.

| $n$ | $p$ | $n$ | $p$ | $n$ | $p$ |
|---|---|---|---|---|---|
| 50 | $\{2, 5\}$ | 500 | $\{2, 5, 10, 25\}$ | 2000 | $\{2, 5, 10, 25\}$ |
| 100 | $\{2, 5, 10\}$ | 1000 | $\{2, 5, 10, 25\}$ | 2500 | $\{2, 5, 10, 25\}$ |
| 250 | $\{2, 5, 10, 25\}$ | 1500 | $\{2, 5, 10, 25\}$ | 3000 | $\{2, 5, 10, 25\}$ |

For each $(n, p)$ pair, we have a total of 6 simulation settings, with $R^2 = 0.2, 0.5, 0.8$ and individual causal effect zero or non-zero, as described in the main body of the paper, to study and compare the estimation and inference results. As for the comparison of sampling times, we only keep one setting of each $(n, p)$ pair, i.e., $R^2 = 0.5$ and non-zero effect, as changes in $R^2$ and the effect would not essentially affect the simulation speed and therefore one setting would already be sufficient in illustrating the acceleration performance of our proposed method PSRSRR.

## B.2 HYPERPARAMETER TEMPERATURE ANALYSIS

To investigate the effect of the hyperparameter $T$ on our algorithm performance, we conduct the analysis by changing the value of $T$ under different pairs of $(n, p)$ with $R^2 = 0.5$ and non-zero effects. Across all simulation settings, we varied the temperature $T$ in $\{0.01, 0.1, 0.5, 0.9, 0.99\}$ with 10 repetitions for each case, and both the MSE and sampling time were recorded for comparison.

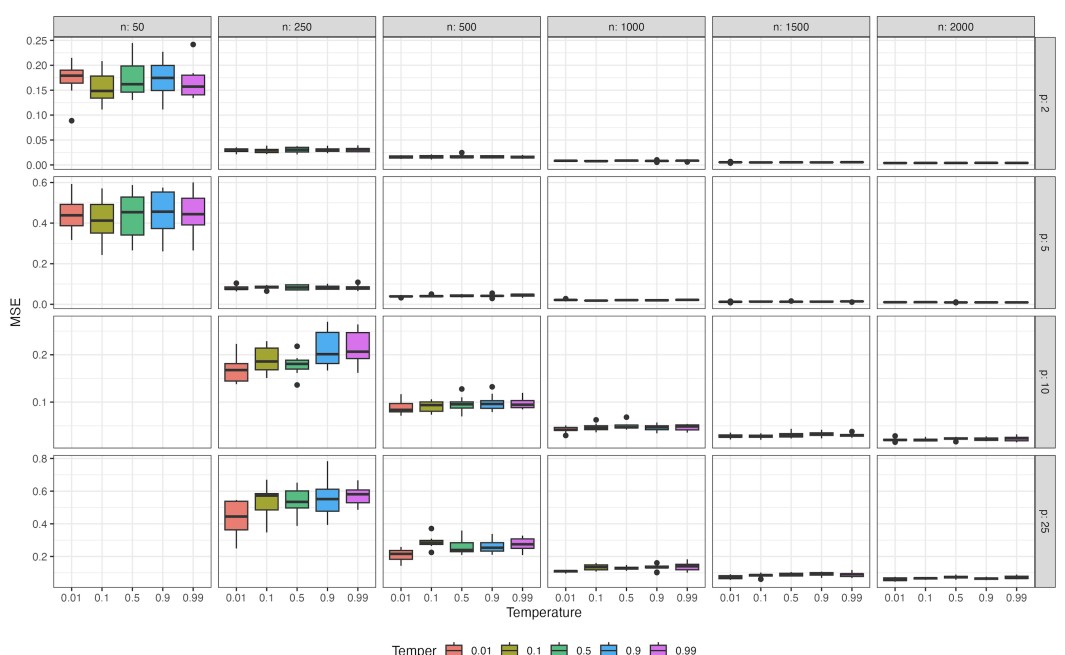

Figure 4: Temperature Analysis: Comparison of MSE

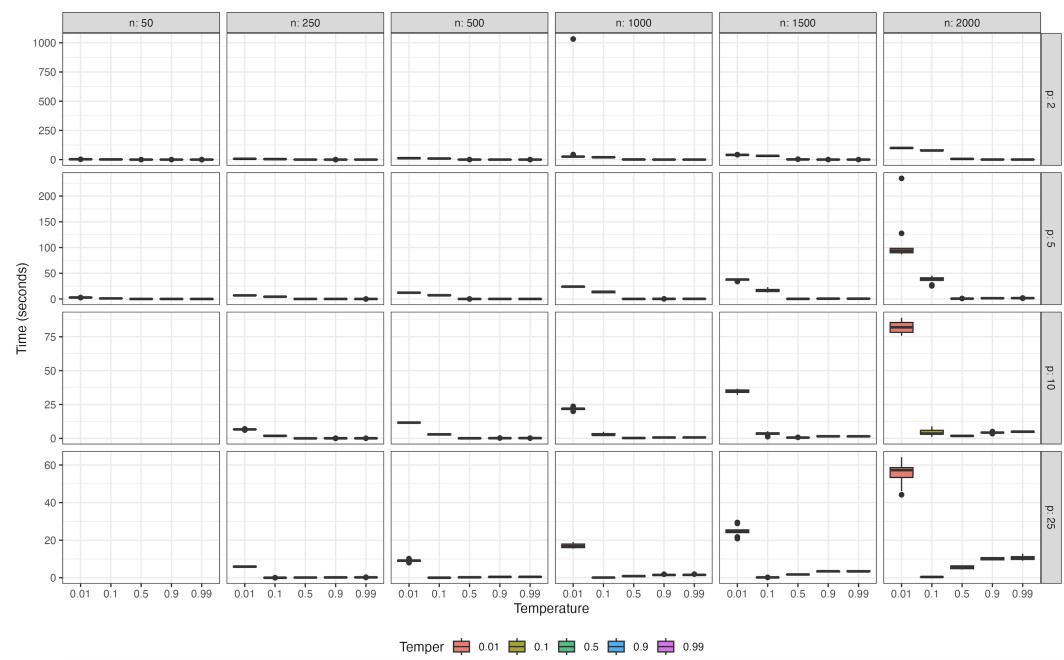

Figure 5: Temperature Analysis: Comparison of Sampling Time

According to Figure 4, we can conclude that $T$ has no significant influence on the MSE of the estimation algorithm, with smaller $T$ performing slightly better in small-sample, high-dimensional settings (e.g. $n = 250$ and $p \in \{10, 25\}$ ). Figure 5 shows that very small temperatures (especially $T = 0.01$) can dramatically increase the sampling time when $n$ is large, while this increase tends to decrease when $p$ gets larger with fixed $n$ (note that the time axis differs in each row). And it is also noticeable that the optimal $T$ moves to 0 when $p$ increases.

Therefore, combined with the above observation, our empirical choice $T = 1.8/p$ lands in a stable and favorable position with these trends under different $(n, p)$, which effectively moves to smaller values when $p$ increases, resulting in better MSE and shorter sampling times. This choice also reflects the intuition that the increase in $p$ leads to a generally smaller change in $M(\mathbf{W})$ after a single pair-switch and it is thus reasonable for the algorithm to favor less exploration.

### B.3 UNIFORMITY VERIFICATION

#### B.3.1 KOLMOGOROV-SMIRNOV TEST FOR $H_0 : M(\mathbf{W}_{\text{PSRSRR}}) \overset{d}{=} M(\mathbf{W}_{\text{RR}})$

Besides the non-direct verification of near-uniformity of the assignment distribution generated by our proposed method PSRSRR, we here verify more directly whether the assignments generated by PSRSRR can be regarded as uniformly distributed and possess better uniformity than PSRR.

The verification procedure is designed as follows. For each setting, we generate $10,000$ assignments using PSRSRR, PSRR and RR, respectively. We calculate the Mahalanobis distance of each assignment and obtain the Mahalanobis distance distribution. We conduct one Kolmogorov-Smirnov test on whether the Mahalanobis distance of assignments generated by PSRSRR and that generated by RR are the same, and another Kolmogorov-Smirnov test on whether the Mahalanobis distance of assignments generated by PSRR and that generated by RR are the same. This way, we obtain one $p$-value for each hypothesis testing, which has the implication whether the accelerated algorithm could generate assignments that have the distribution to be accepted as the same as the assignment distribution generated by RR in the sense of Mahalanobis distance. We repeat this process for $100$ times to get an empirical distribution of $p$-values, which can be visualized in a boxplot.

Due to the computational constraint of RR, we limit our verification to some small-sample settings, and also the threshold selection strategy to $p_a$ only, since $\nu_{p,a}$-based strategy would likely lead

to much stricter criteria that RR could hardly handle. We present the boxplots obtained from the verification procedures described above as follows. In all these settings, PSRSRR shows a great proportion of $p$-values above the rejection threshold $0.05$, indicating a good alignment with RR, verifying the good uniformity of PSRSRR in the sense of Mahalanobis distance. While PSRR, especially in (a) and (d), appears to have many $p$-values below the $0.05$ threshold, meaning that in many of the repetitions, the null hypothesis that the Mahalanobis distance distribution of the assignments generated by PSRR is the same as that generated by RR is rejected. Therefore, we can conclude that PSRSRR can generally be considered as a better approximation of RR than PSRR.

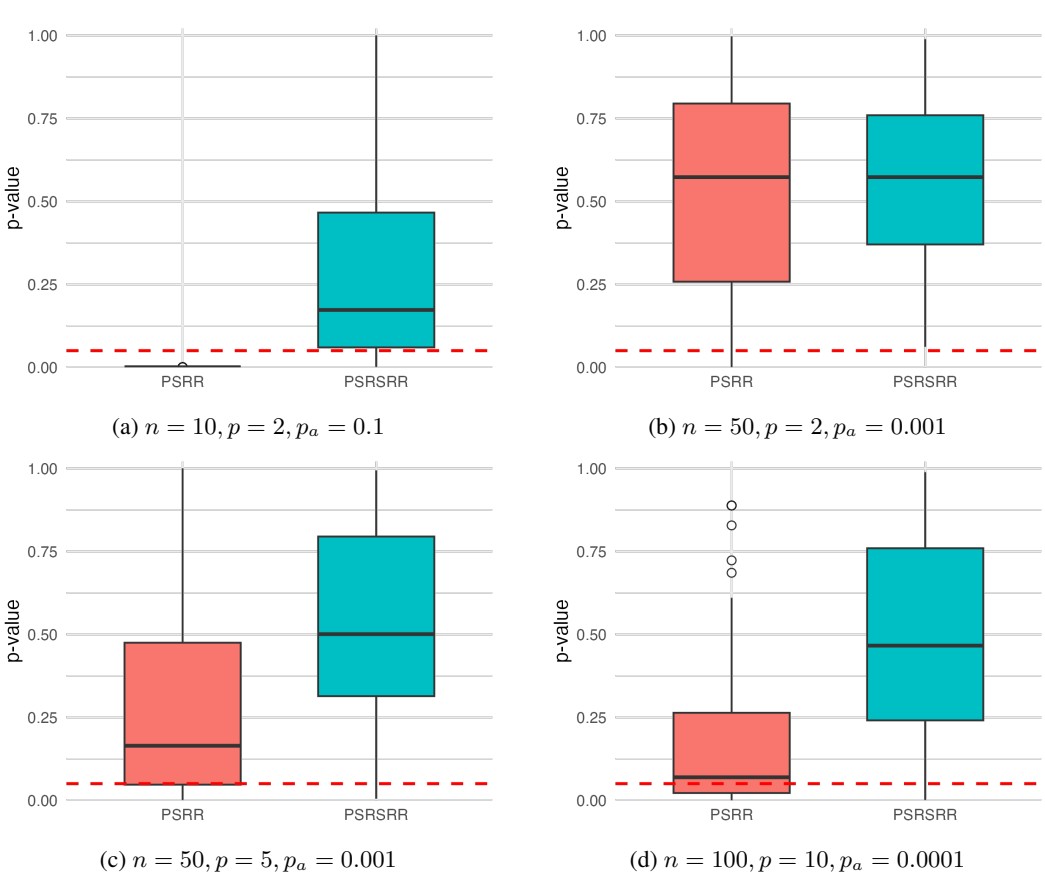

(a) $n = 10, p = 2, p_a = 0.1$

(b) $n = 50, p = 2, p_a = 0.001$

(c) $n = 50, p = 5, p_a = 0.001$

(d) $n = 100, p = 10, p_a = 0.0001$

Figure 6: Boxplots of Kolmogorov-Smirnov test $p$-value distribution. In each figure, the left boxplot is for $H_0 : M(\mathbf{W}_{\text{PSRR}}) \overset{d}{=} M(\mathbf{W}_{\text{RR}})$ and the right one is for $H_0 : M(\mathbf{W}_{\text{PSRSRR}}) \overset{d}{=} M(\mathbf{W}_{\text{RR}})$. The red line indicates the $0.05$ significance threshold of rejecting the null hypothesis.

### B.3.2 KOLMOGOROV-SMIRNOV TEST FOR $H_0 : M(\mathbf{W}_{\text{PSRSRR}}) \stackrel{d}{=}$ TRUNCATED CHI-SQUARE

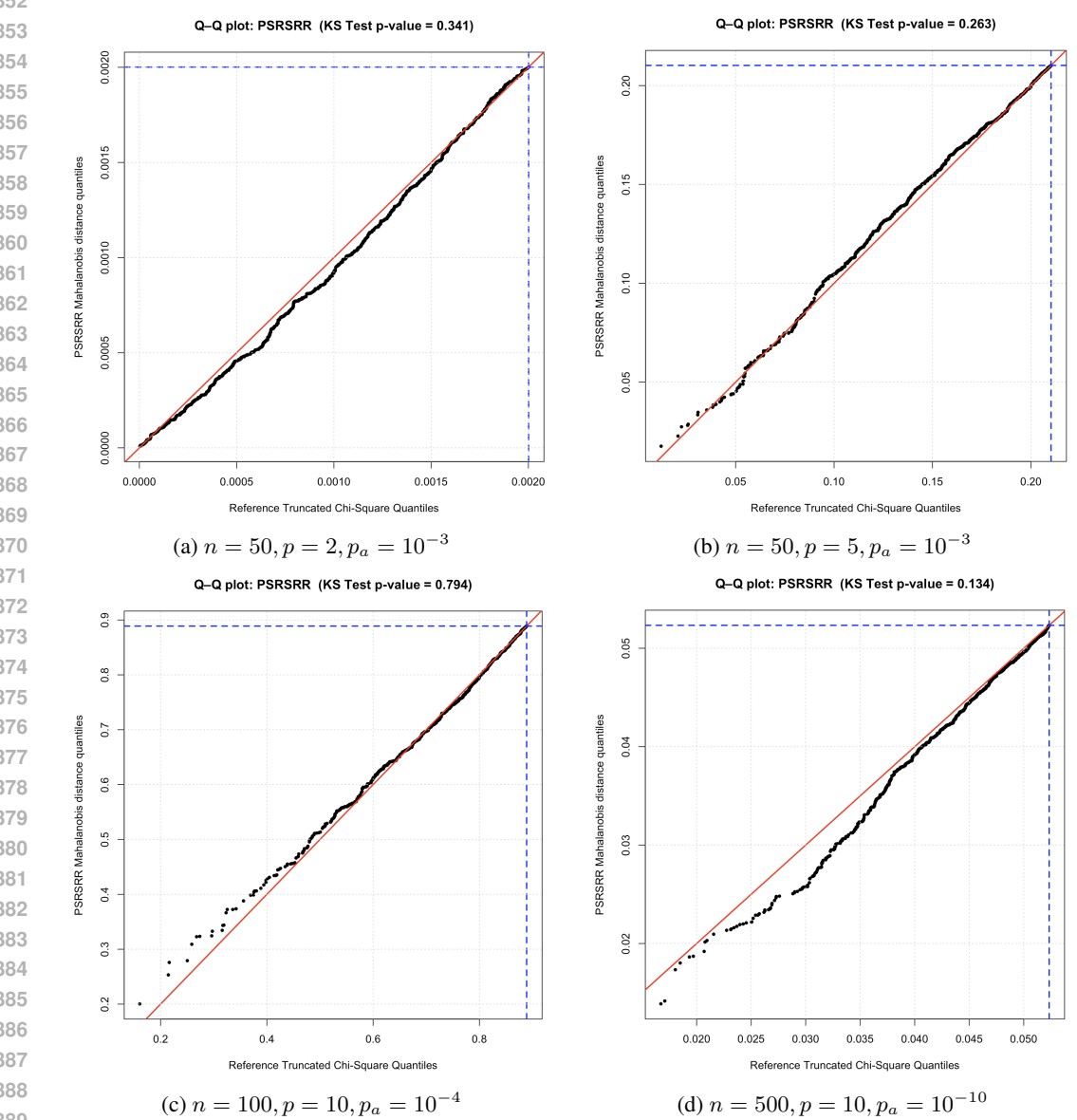

(a) $n = 50, p = 2, p_a = 10^{-3}$

(b) $n = 50, p = 5, p_a = 10^{-3}$

(c) $n = 100, p = 10, p_a = 10^{-4}$

(d) $n = 500, p = 10, p_a = 10^{-10}$

Figure 7: Quantile-quantile plots with Kolmogorov-Smirnov test $p$-values. The red line indicates the 45-degree line (perfect equality) and the blue lines for the threshold value $a$.

To facilitate direct uniformity testing in a more efficient way, we take advantage of the result given in Morgan & Rubin (2012) that the asymptotic acceptance probability of the treatment assignment is $p_a = \mathbb{P}\left(\chi_p^2 \leq a\right)$, and therefore we compare the Mahalanobis distances of our assignments with a truncated chi-square distribution $M_{\text{asym}} \sim \chi_p^2|\left(\chi_p^2 \leq a\right)$ via a two-sample Kolmogorov-Smirnov test. This diagnostic tool is easy to implement and prevents the trouble of generating assignments using RR as a baseline, thus greatly improving computational efficiency as well as providing solid theoretical guarantee.

We attach the quantile-quantile plots in Figure 7 and 8, with the x-axis representing the simulated values of the corresponding truncated chi-square distribution, and the y-axis representing the sorted Mahalanobis distance values generated by our proposed algorithm PSRSRR. Since the distribution is asymptotic, we include results from relatively large sample sizes, and choose $p_a$ to be relatively

small, as when $p$ increases, one of our previous threshold selection criteria following the practical implications in Wang & Li (2022), could result in very small $p_a$ values (e.g., when $\nu_{p,a}$ is set as 0.01, we obtain $p_a$ of order $10^{-9}$ for $p = 10$ and $10^{-21}$ for $p = 25$). For each Q-Q plot, we also include the p-value from the Kolmogorov-Smirnov test for equality of distributions. We conclude that the assignments generated by our proposed algorithm PSRSRR exhibit good uniformity, thereby supporting the use of theoretical results derived under uniform sampling from the acceptance space.

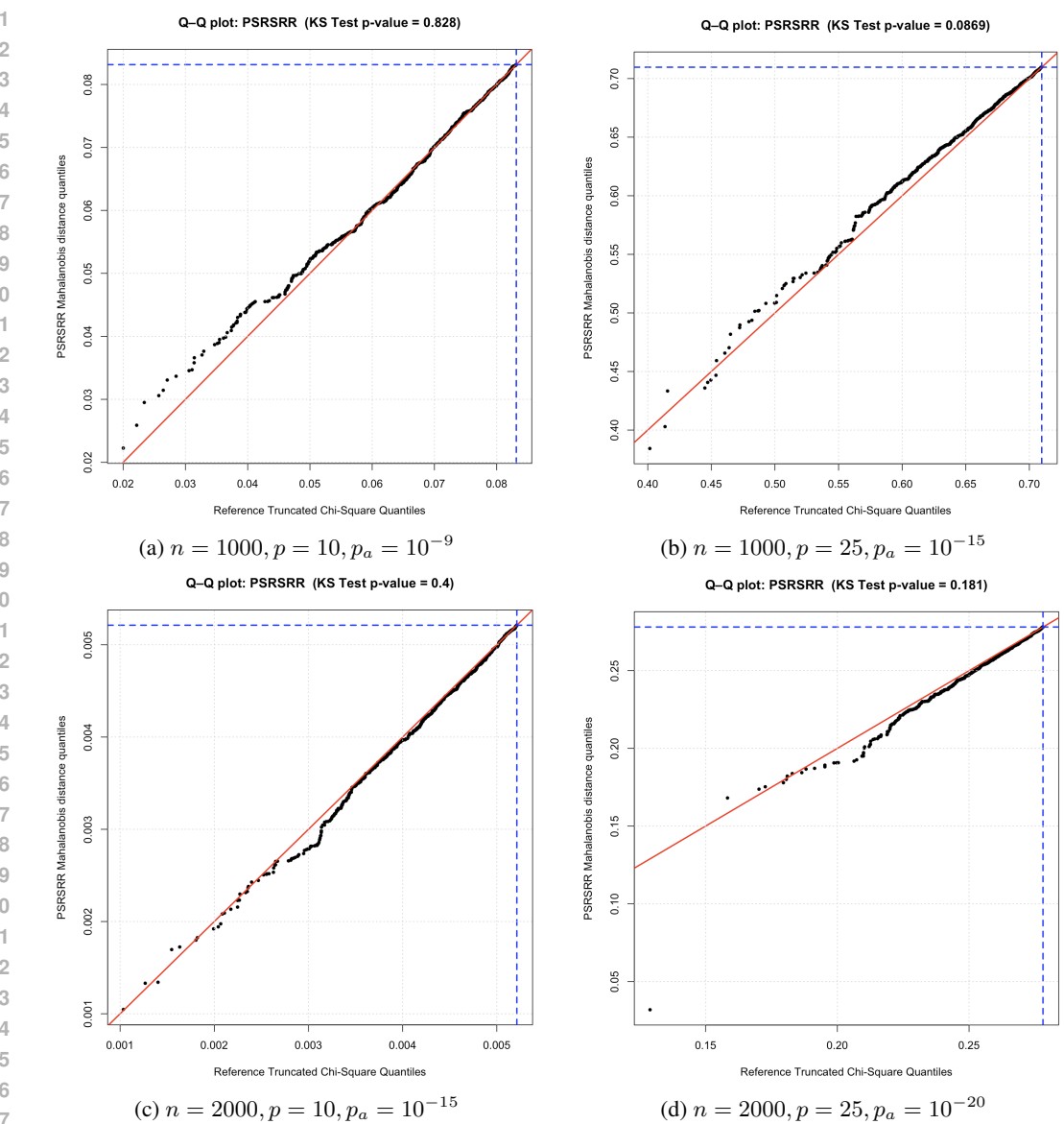

(a) $n = 1000, p = 10, p_a = 10^{-9}$

(b) $n = 1000, p = 25, p_a = 10^{-15}$

(c) $n = 2000, p = 10, p_a = 10^{-15}$

(d) $n = 2000, p = 25, p_a = 10^{-20}$

Figure 8: Quantile-quantile plots with Kolmogorov-Smirnov test $p$-values (cont'd). The red line indicates the 45-degree line (perfect equality) and the blue lines for the threshold value $a$.

### B.4 ESTIMATION AND INFERENCE RESULTS FOR ADDITIONAL SETTINGS

In the main body of the paper, we have already presented and discussed results under the simulation settings with $R^2 = 0.5$. Here, we include more simulation results under settings with $R^2 = 0.2$ and $R^2 = 0.8$, in order to provide a more comprehensive overview of the performance of our proposed algorithm PSRSRR.

In the following, we present the comparison plots of MSE for all three $R^2$ settings, as well as the comparative statistics regarding the MSE improvement of PSRSRR with respect to other competing methods. We can obtain the similar conclusion as given in the main body of the paper, that PSRSRR has the best performance among all methods, with very similar estimation results as GSW. And when the number of covariates $p$ increases, i.e., in more completed simulation settings, the improvement of PSRSRR compared with PSRR and CR is more stable and more significant. And when $R^2$ increases, i.e., the covariates are more explanatory and there is less noise, the improvement of PSRSRR becomes more significant (except for GSW). Therefore, MSE comparison results show that PSRSRR is superior, especially in large-sample and high-dimensional settings.

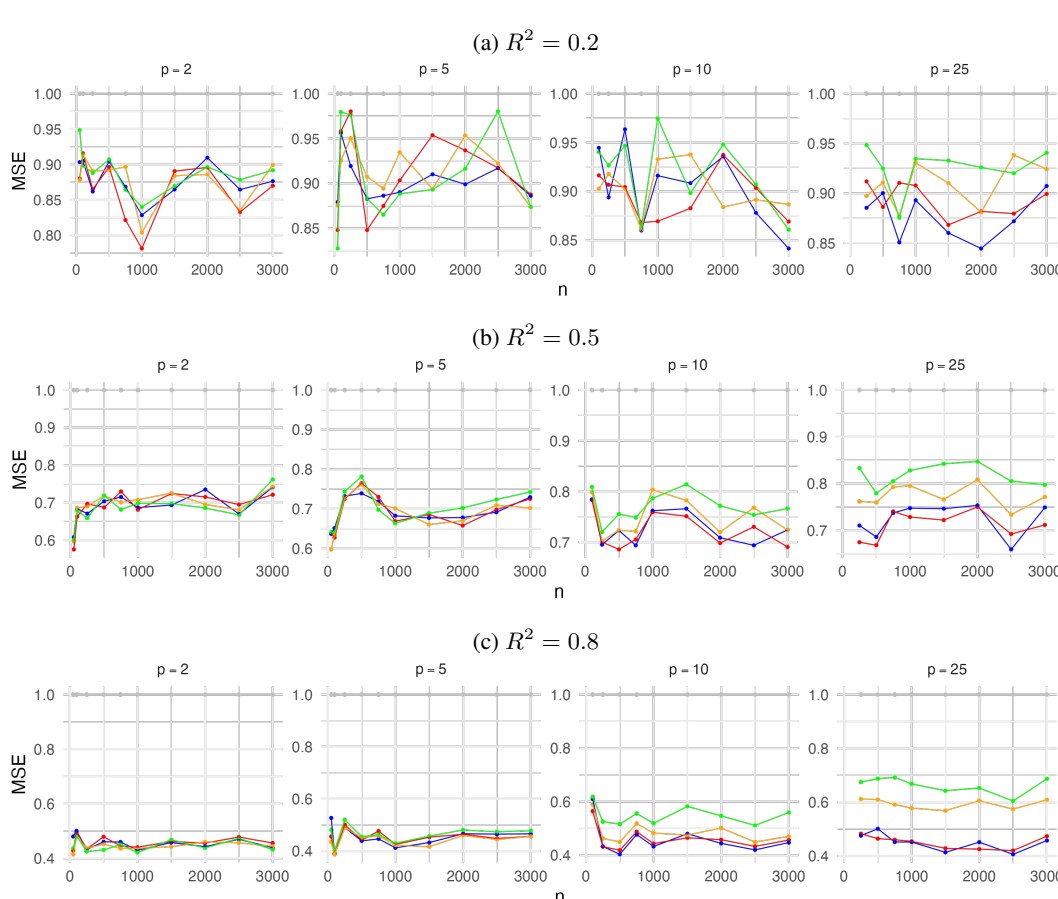

Figure 9: Comparison plots of MSE relative to CR, by sample size $n$ and number of covariates $p$.

| $R^2$ | Statistic | Relative MSE Ratio | | | |
| --- | --- | --- | --- | --- | --- |
| | | To CR | To RR | To PSRR | To GSW |
| 0.2 | Range | $[78\%, 98\%]$ | $[89\%, 107\%]$ | $[92\%, 107\%]$ | $[94\%, 107\%]$ |
| | Mean | $89\%$ | $98\%$ | $99\%$ | $100\%$ |
| 0.5 | Range | $[56\%, 78\%]$ | $[81\%, 107\%]$ | $[88\%, 107\%]$ | $[95\%, 105\%]$ |
| | Mean | $70\%$ | $96\%$ | $97\%$ | $99\%$ |
| 0.8 | Range | $[39\%, 56\%]$ | $[65\%, 111\%]$ | $[70\%, 109\%]$ | $[87\%, 107\%]$ |
| | Mean | $46\%$ | $89\%$ | $95\%$ | $100\%$ |

Table 3: Relative MSE ratios: range and mean of PSRSRR compared with each competing method under different $R^2$.

In the following, we present the comparison plots of confidence interval length for all three $R^2$ settings, as well as the comparative statistics regarding the improvement of PSRSRR with respect to other competing methods. We can obtain similar conclusions that PSRSRR has consistently the best performance among all methods. And its superiority becomes more significant as $R^2$ increases, $n$ increases and $p$ increases, illustrating its strength in large-sample and high-dimensional settings.

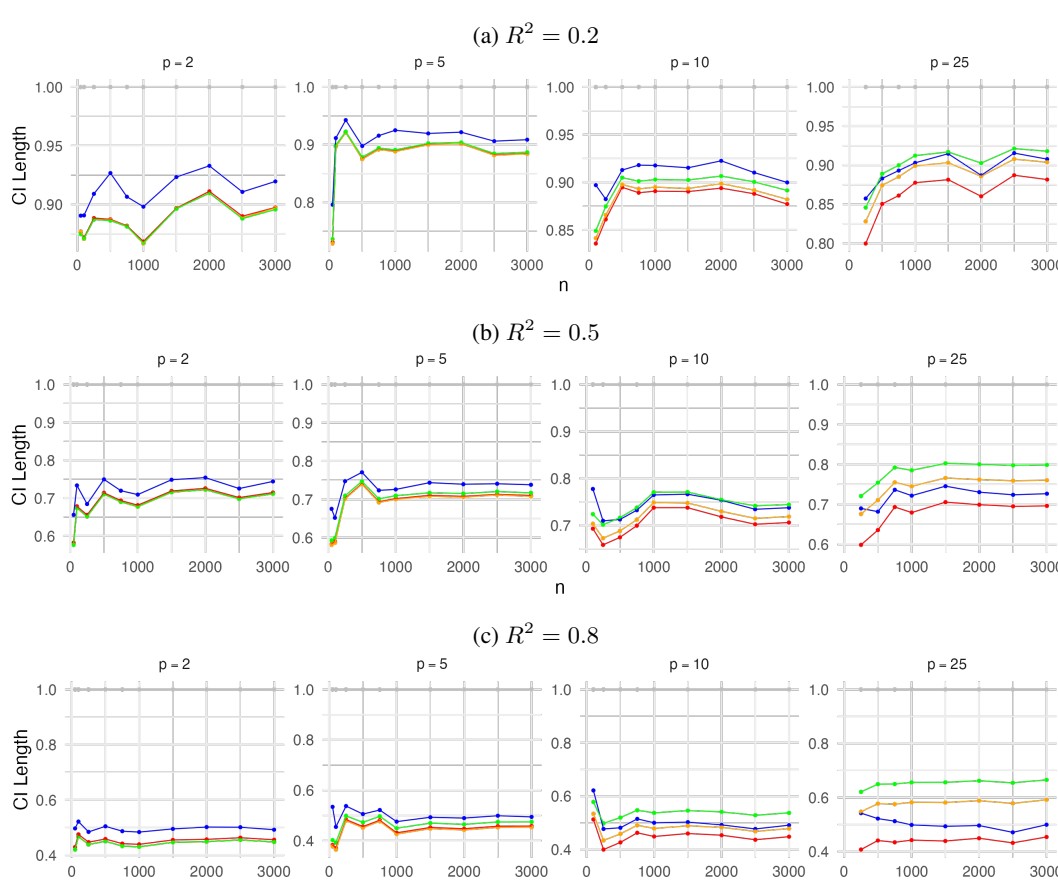

Figure 10: Comparison plots of CI length relative to CR, by sample size $n$ and number of covariates $p$.

| $R^2$ | Statistic | Relative CI Length Ratio | | | |
| --- | --- | --- | --- | --- | --- |
| | | To CR | To RR | To PSRR | To GSW |
| 0.2 | Range | $[73\%, 92\%]$ | $[95\%, 100\%]$ | $[97\%, 100\%]$ | $[92\%, 98\%]$ |
| | Mean | $88\%$ | $99\%$ | $99\%$ | $97\%$ |
| 0.5 | Range | $[58\%, 74\%]$ | $[83\%, 101\%]$ | $[88\%, 101\%]$ | $[86\%, 97\%]$ |
| | Mean | $69\%$ | $96\%$ | $98\%$ | $94\%$ |
| 0.8 | Range | $[37\%, 51\%]$ | $[66\%, 102\%]$ | $[74\%, 102\%]$ | $[72\%, 93\%]$ |
| | Mean | $45\%$ | $88\%$ | $94\%$ | $89\%$ |

Table 4: Relative CI Length ratios: range and mean of PSRSRR compared with each competing method under different $R^2$.

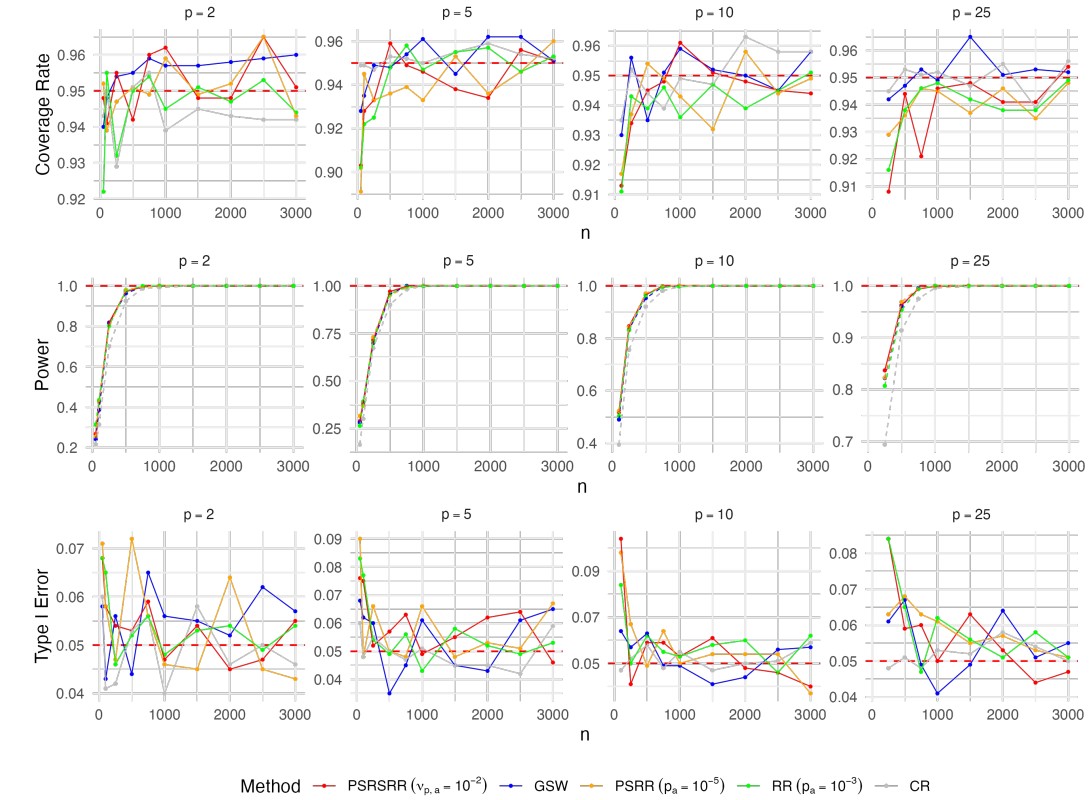

Figure 11: Comparison plots of Coverage Rate, Power and Type I Error under $R^2 = 0.2$.

The above plots are the comparison plots under the settings with $R^2 = 0.2$. We can tell from the Coverage Rate one that PSRSRR can achieve a satisfactory coverage rate, with values closely around the nominal level $95\%$. And PSRSRR has relatively better power when the number of covairates $p$ is larger. As for the Type I Error, when sample size $n$ increases, PSRSRR has error values slightly above or below the error bound $0.05$. These results all show the validity of applying the theoretical results with the ground in uniformity of distribution to PSRSRR, thus non-directly verifying the near-uniformity of the distribution of the assignments generated by our proposed algorithm.

We can obtain similar conclusions from the following plots under the settings with $R^2 = 0.8$. In the Coverage Rate figure, GSW seems to be well above the nominal level $95\%$, indicating the fact that inference of GSW under settings with $R^2 = 0.8$ may tend to be conservative; while PSRSRR still has values closely around the nominal level, showing good inference performance. In Power, compared with settings under $R^2 = 0.2$ and $R^2 = 0.5$, all methods appear to have higher powers, due to the fact that the covariates can explain more variation in the simulation model. And among all methods, PSRSRR has an obviously higher power when $p$ is relatively large. As for the Type I Error, when sample size $n$ increases, PSRSRR has error values reasonably around $0.05$.

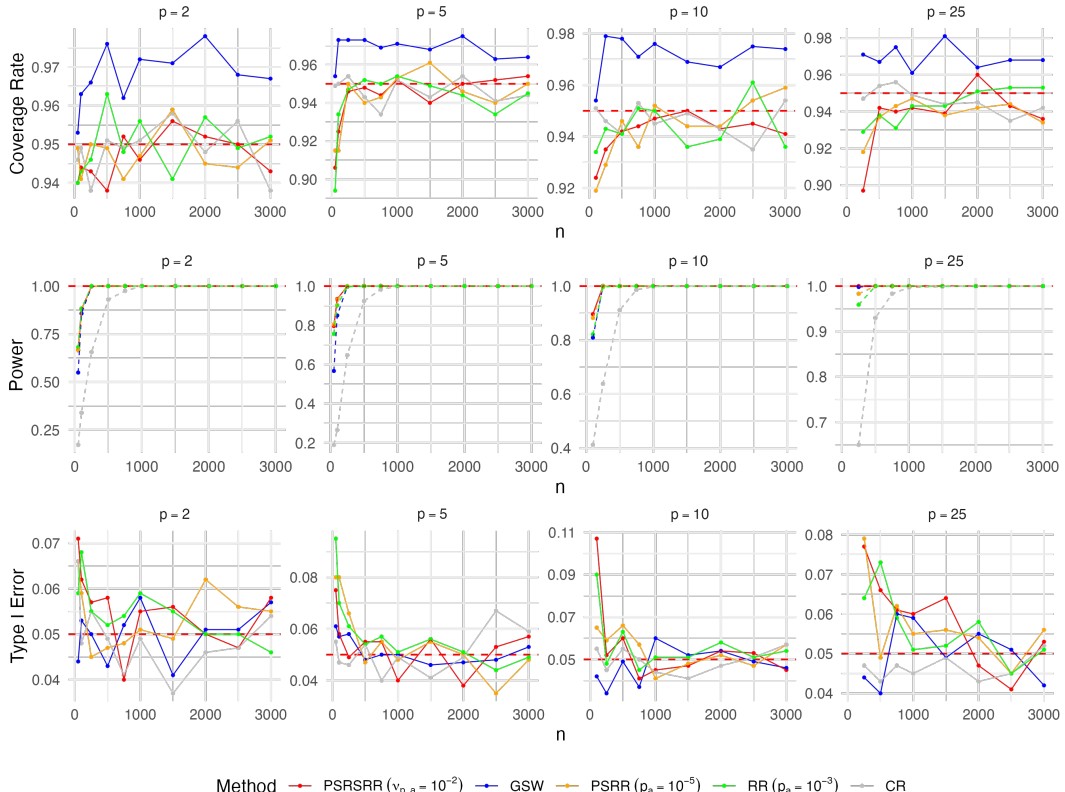

Figure 12: Comparison plots of Coverage Rate, Power and Type I Error under $R^2 = 0.8$.

## B.5 EXTENSION TO COVARIATE-PRIORITIZED RERANDOMIZATION METHODS: ReB AS AN EXAMPLE

### B.5.1 THEORETICAL ARGUMENT

As pointed out in Liu et al. (2025), basic rerandomization methods simply assign uniform weights in the covariates and fail to prioritize covariates that are believed to be more important and strongly associated with the potential outcomes.

This line of research, which prioritizes balancing important covariates, has attracted increasing attention. ReM$_T$ assigns differential weights to covariates by partitioning them into tiers according to their relative importance (Morgan & Rubin, 2015). Ridge-ReM mitigates collinearity among covariates by employing a ridge-modified Mahalanobis distance (Branson & Shao, 2021). PCA-ReM uses rerandomization on the leading $k$ principal components of covariates (Zhang et al., 2024). ReW incorporates prior information on covariate importance through a weighted Euclidean distance used to assess covariate balance (Lu et al., 2023). ReB introduces a Bayesian criterion for rerandomization by encoding prior knowledge on the covariates' importance via a prior distribution (Liu et al., 2025).

In this subsection, we argue that, although the main theoretical derivation and algorithm designs in this paper are proposed using Mahalanobis distance, our results can be readily incorporated into

many of the above mentioned covariate-prioritized rerandomization methods, via a change of covariate imbalance metric. Note that our main Algorithms 2, 3 and Theorems 1, 2 only use the Mahalanobis distance $M(\mathbf{W})$ as a general metric notation while do not depend on the specific formula of this metric. In addition, the choice of threshold $a$ is very flexible and could be chosen according to the different requirements of different methods. We can thus adopt a distance metric that assigns different weights to the covariate balance to achieve the same goal of prioritizing important covariates. As follows, we have explicitly written out the distance metric $M(\mathbf{W})$ we need to use in order to accommodate some above mentioned methods. The notation is consistent with those introduced in Section 2.

- Branson & Shao (2021) Equation (6):

$$M_\lambda = \left(\overline{\mathbf{X}}_t - \overline{\mathbf{X}}_c\right)^\top \left[(\mathrm{Cov}\left(\overline{\mathbf{X}}_t - \overline{\mathbf{X}}_c\right) + \lambda \mathbf{I}_p\right]^{-1} \left(\overline{\mathbf{X}}_t - \overline{\mathbf{X}}_c\right) \tag{10}$$

  for some prespecified regularization parameter $\lambda \geq 0$ and $\mathbf{I}_p$ denotes the $p$-dimensional identity matrix.

- Zhang et al. (2024) Equation (6): Define $\mathbf{X} = \mathbf{U}\mathbf{D}\mathbf{V}^\top$ to be the singular value decomposition of $\mathbf{X}$, where $\mathbf{U} \in \mathbb{R}^{n \times d}$ and $\mathbf{V} \in \mathbb{R}^{p \times d}$ correspond to the matrices of the left and right singular vectors, with $\mathbf{U}^\top\mathbf{U} = \mathbf{V}^\top\mathbf{V} = \mathbf{I}_d$ and $d = \min\{n, p\}, \mathbf{D} = \mathrm{diag}\{\sigma_1, \ldots, \sigma_d\}$ is a diagonal matrix composed of non-negative singular values $\sigma_1 \geq \cdots \geq \sigma_d > 0$. Without loss of generality, let $d = p$ (i.e., $n > p$) so that $\mathbf{Z} = (z_{ij}) = \mathbf{U}\mathbf{D}$ are the principal components of $\mathbf{X}$. Let $\mathbf{Z}_k = \mathbf{U}_k\mathbf{D}_k = \left(\mathbf{z}_1^{(k)}, \ldots, \mathbf{z}_n^{(k)}\right)^\top \in \mathbb{R}^{n \times k}$ denote the matrix of the top $k(k \leq p)$ principal components of $\mathbf{X}$, where $\mathbf{z}_i^{(k)} \in \mathbb{R}^{k \times 1}$ contains the first $k$ elements of the $i$-th row of $\mathbf{Z}$, and $\mathbf{U}_k \in \mathbb{R}^{n \times k}$ is the first $k$ columns of $\mathbf{U}$ and $\mathbf{D}_k \in \mathbb{R}^{k \times k}$ is the top $k$-dimensional submatrix of $\mathbf{D}$. Similarly, let $\tilde{\mathbf{Z}}_{p-k} = \tilde{\mathbf{U}}_{p-k}\tilde{\mathbf{D}}_{p-k} \in \mathbb{R}^{n \times (p-k)}$ denote the last $p - k$ principal components, where $\tilde{\mathbf{U}}_{p-k} \in \mathbb{R}^{n \times (p-k)}$ is the last $p - k$ columns of $\mathbf{U}$ and $\tilde{\mathbf{D}}_k \in \mathbb{R}^{(p-k) \times (p-k)}$ is the corresponding $(p - k)$-dimensional submatrix of $\mathbf{D}$. Mahalanobis distance is calculated based on the top $k$ principal components,

$$\boldsymbol{M}_k = \left(\overline{\mathbf{z}}_T^{(k)} - \overline{\mathbf{z}}_C^{(k)}\right)^\top \boldsymbol{\Sigma}_z^{-1} \left(\overline{\mathbf{z}}_T^{(k)} - \overline{\mathbf{z}}_C^{(k)}\right), \tag{11}$$

  where $\overline{\mathbf{z}}_T^{(k)}$ and $\overline{\mathbf{z}}_C^{(k)}$ are defined as

$$\overline{\mathbf{z}}_T^{(k)} = \frac{2}{n}\mathbf{Z}_k^\top \mathbf{W} \quad \text{and} \quad \overline{\mathbf{z}}_C^{(k)} = \frac{2}{n}\mathbf{Z}_k^\top (\mathbf{1}_n - \mathbf{W}),$$

  and $\boldsymbol{\Sigma}_z = C_n \mathbf{Z}_k^\top \mathbf{Z}_k = C_n \mathbf{D}_k^2$ with $C_n = 4/(n^2 - n)$ and $\mathbf{1}_n$ is the $n$-dimensional all-ones vector.

- Liu et al. (2025) Theorem 1 ($\beta$ is known):Denote $\boldsymbol{\beta} = \boldsymbol{V}_{xx}^{-1}\boldsymbol{V}_{x\tau}$, where

$$\boldsymbol{V}_{xx} = \frac{n^2}{n_t n_c(n-1)} \sum_{i=1}^n \left(\mathbf{X}_i - \overline{\mathbf{X}}\right)\left(\mathbf{X}_i - \overline{\mathbf{X}}\right)^\top \qquad \boldsymbol{V}_{x\tau} = \frac{n}{n_t}S_{Y(1),\mathbf{X}}^2 + \frac{n}{n_c}S_{Y(0),\mathbf{X}}^2$$

$$\boldsymbol{S}_{Y(w),\boldsymbol{X}}^2 = \frac{1}{n-1} \sum_{i=1}^n \left(Y_i(w) - \frac{1}{n}\sum_{i=1}^n Y_i(w)\right)\left(\mathbf{X}_i - \overline{\mathbf{X}}\right)^\top, \; w \in \{0, 1\}$$

  The distance metric is calculated as

$$M_\beta = \left[\sqrt{n}\left(\overline{\mathbf{X}}_t - \overline{\mathbf{X}}_c\right)\right]^\top \boldsymbol{\beta}\boldsymbol{\beta}^\top \left[\sqrt{n}\left(\overline{\mathbf{X}}_t - \overline{\mathbf{X}}_c\right)\right] \tag{12}$$

  with threshold $a_\beta = \boldsymbol{V}_{\tau x}\boldsymbol{V}_{xx}^{-1}\boldsymbol{V}_{x\tau}\xi_{p_a}$, where $\xi_{p_a}$ is the $p_a$-quantile of the $\chi_1^2$ distribution.

Therefore, our proposed theories and algorithms do not only have limited contribution to the rerandomization literature using the Mahalanobis distance but also serve as a general framework that could well accommodate and significantly reduce computational costs of many of the methods that prioritize important covariates.

### B.5.2 Proof of Distance Metric Calculation Trick

To implement our algorithm combined with non-uniform covariate weighting as shown in Liu et al. (2025), we first perform a similar derivation for the calculation trick of pair switching in Zhu & Liu (2023) when using the metric 12.

**Lemma 10** *Let $M_\beta^{(t)}$ denote the distance metric of assignment $\mathbf{W}^{(t)}$ at time step $t$ calculated using Equation 12, and $i, j$ denote the pair that is examined for conducting pair-switching with $W_i^{(t)} = 1$ and $W_j^{(t)} = 0$. Denote $\mathbf{W}^*$ the assignment vector after pair-switching $i, j$ elements, then the distance metric calculation for one pair-switching in assignments could be simplified as follows,*

$$M_\beta(\mathbf{W}^*) = M_\beta^{(t)} - \left(2\sum_{l=1}^n W_l^{(t)} H_{il} - H_{ii}\right) + \left(2\sum_{l=1}^n W_l^* H_{jl} - H_{jj}\right) + h_i - h_j \qquad (13)$$

*where $\mathbf{H} = (H_{ij}) = \frac{n^3}{n_t^2 n_c^2}\mathbf{X}\mathbf{V}_{xx}^{-1}\mathbf{V}_{x\tau}\mathbf{V}_{\tau x}\mathbf{V}_{xx}^{-1}\mathbf{X}^\top, \mathbf{h} = \frac{2n_t}{n}\mathbf{H}\cdot\mathbf{1}_n$ with $\mathbf{1}_n$ as an $n$-dimensional column vector of 1's.*

We prove Lemma 10 below. Notice that

$$\overline{\mathbf{X}}_t - \overline{\mathbf{X}}_c = \frac{\mathbf{X}^\top\mathbf{W}}{n_t} - \frac{\mathbf{X}^\top(\mathbf{1}_n - \mathbf{W})}{n_c} = \frac{n}{n_t n_c}\mathbf{X}^\top\left(\mathbf{W} - \frac{n_t}{n}\mathbf{1}_n\right) \qquad (14)$$

Plug this into the distance metric 12, we have

$$
\begin{aligned}
M_\beta(\mathbf{W}) &= \left[\sqrt{n}\left(\overline{\mathbf{X}}_t - \overline{\mathbf{X}}_c\right)\right]^\top \boldsymbol{\beta}\boldsymbol{\beta}^\top \left[\sqrt{n}\left(\overline{\mathbf{X}}_t - \overline{\mathbf{X}}_c\right)\right] \\
&= n\left[\frac{n}{n_t n_c}\mathbf{X}^\top\left(\mathbf{W} - \frac{n_t}{n}\mathbf{1}_n\right)\right]^\top \mathbf{V}_{xx}^{-1}\mathbf{V}_{x\tau}\left(\mathbf{V}_{xx}^{-1}\mathbf{V}_{x\tau}\right)^\top\left[\frac{n}{n_t n_c}\mathbf{X}^\top\left(\mathbf{W} - \frac{n_t}{n}\mathbf{1}_n\right)\right] \\
&= \left(\mathbf{W} - \frac{n_t}{n}\mathbf{1}_n\right)^\top\left(\frac{n^3}{n_t^2 n_c^2}\mathbf{X}\mathbf{V}_{xx}^{-1}\mathbf{V}_{x\tau}\mathbf{V}_{\tau x}\mathbf{V}_{xx}^{-1}\mathbf{X}^\top\right)\left(\mathbf{W} - \frac{n_t}{n}\mathbf{1}_n\right) \\
&= \left(\mathbf{W} - \frac{n_t}{n}\mathbf{1}_n\right)^\top \mathbf{H}\left(\mathbf{W} - \frac{n_t}{n}\mathbf{1}_n\right) = \mathbf{W}^\top\mathbf{H}\mathbf{W} - \mathbf{W}\cdot\mathbf{h} + \frac{n_t^2}{n^2}\mathbf{1}_n^\top\cdot\mathbf{H}\cdot\mathbf{1}_n
\end{aligned}
$$
$$(15)$$

With $W_i^{(t)} = 1$ and $W_j^{(t)} = 0$, and $W_i^* = 0$ and $W_j^* = 1$, the difference of the distance metrics can be calculated as

$$
\begin{aligned}
M_\beta(\mathbf{W}^*) - M_\beta^{(t)} &= \left(\mathbf{W}^{*\top}\mathbf{H}\mathbf{W}^* - \mathbf{W}^*\cdot\mathbf{h}\right) - \left(\mathbf{W}^{(t)\top}\mathbf{H}\mathbf{W}^{(t)} - \mathbf{W}^{(t)}\cdot\mathbf{h}\right) \\
&= \left(\mathbf{W}^{*\top}\mathbf{H}\mathbf{W}^* - \mathbf{W}^{(t)\top}\mathbf{H}\mathbf{W}^{(t)}\right) - \left(\mathbf{W}^*\cdot\mathbf{h} - \mathbf{W}^{(t)}\cdot\mathbf{h}\right) \\
&= \left(2\sum_{l\neq j} W_l^* H_{jl} + H_{jj}\right) - \left(2\sum_{l\neq i} W_l^{(t)} H_{il} + H_{ii}\right) + h_i - h_j \\
&= \left(2\sum_{l=1}^n W_l^* H_{jl} - H_{jj}\right) - \left(2\sum_{l=1}^n W_l^{(t)} H_{il} - H_{ii}\right) + h_i - h_j
\end{aligned}
$$
$$(16)$$

which concludes the proof. $\qquad\square$

### B.5.3 Experiments

Below, we follow the weighted metric equation 12 used in Liu et al. (2025) as an example, to conduct simulations to illustrate that our algorithm could be easily extended to accommodate the method, and show this accommodation could lead to great computational efficiency compared with the original rejection sampling technique used in Liu et al. (2025).

We adopt the data generation mechanism proposed in Liu et al. (2025), whose parameter explanation is attached in Table 5, and note that we specifically focus on settings with $\sigma_b^2 = 1$, since under

their mechanism where $Y_i(1) = 5 + \boldsymbol{b}_1^\top \boldsymbol{X}_i + \epsilon_{1,i}, Y_i(0) = \boldsymbol{b}_0^\top \boldsymbol{X}_i + \epsilon_{0,i}, i = 1, \ldots, n$, covariates $\boldsymbol{X} \sim \mathcal{N}\left(\boldsymbol{0}_p, (1-\rho)\boldsymbol{I}_p + \rho\boldsymbol{1}_p\boldsymbol{1}_p^\top\right), \epsilon_1, \epsilon_0 \sim \mathcal{N}\left(0, \sigma_\epsilon^2\right)$ where $\sigma_\epsilon^2$ is properly set to tune the average value of noise level $R^2$ as expected, and regression coefficients $\boldsymbol{b}_1 \sim \mathcal{N}\left(2 \cdot \boldsymbol{1}_p, 2\sigma_{\boldsymbol{b}}^2 \cdot \boldsymbol{I}_p\right), \boldsymbol{b}_0 \sim \mathcal{N}\left(\boldsymbol{1}_p, 2\sigma_{\boldsymbol{b}}^2 \cdot \boldsymbol{I}_p\right)$, setting $\sigma_{\boldsymbol{b}}^2$ would lead to non-uniformity in covariate weights. And for clarity, we have simplified the settings to only vary in the sample size.

Table 5: Parameter description in simulation study.

| Parameter | Levels | Description |
|---|---|---|
| $n$ | $\{100, 200, 500, 1000\}$ | Total sample size |
| $p$ | $\{5\}$ | Dimensionality of $\mathbf{X}$ |
| $R^2$ | $\{0.5\}$ | Noise level of regression model |
| $\rho$ | $\{0.2\}$ | Correlation coefficient of $\mathbf{X}$ |
| $\sigma_b^2$ | $\{1\}$ | Variance of non-uniformity in covariate weighting |

Our comparison results are summarized in Table 6, where $p_a$ is set as $10^{-5}$ as in previous comparisons, and 1000 treatment assignments are sampled in each simulation. The value achieving the best performance in each setting is shown in **bold**. From the results, we can clearly verify the asymptotic MSE improvement of using the weighted distance metric 12 as proved in Liu et al. (2025) (see comparison between RR and ReB), and at the same time, we can also conclude that our proposed acceleration algorithm is easy to incorporate into the original ReB method, and could result in significant computational cost reduction. We also point out that the MSE values of ReB + PSRSRR are comparably better, which, by further digging into the sampling process in the simulations, can be partially attributed to the fact that ReB + PSRSRR could reach a higher proportion of satisfactory distance metrics when setting the same max iteration numbers in the algorithm implementations.

Table 6: Bias, MSE and sampling time comparison between RR, ReB and ReB + PSRSRR under sample size $n \in \{100, 200, 500, 1000\}$.

| Sample size $n$ | Method | Bias | MSE | Time (s) |
|---|---|---|---|---|
| | RR | 0.064 | 1.378 | 173.143 |
| 100 | ReB | 0.058 | 1.311 | 164.481 |
| | ReB + PSRSRR | **0.003** | **1.015** | **39.601** |
| | RR | **0.003** | 0.542 | 344.730 |
| 200 | ReB | 0.004 | 0.529 | 532.278 |
| | ReB + PSRSRR | -0.005 | **0.421** | **46.219** |
| | RR | -0.015 | 0.268 | 894.383 |
| 500 | ReB | **-0.010** | 0.253 | 912.444 |
| | ReB + PSRSRR | 0.021 | **0.175** | **121.343** |
| | RR | **-0.002** | 0.112 | 1873.277 |
| 1000 | ReB | 0.002 | 0.097 | 1875.811 |
| | ReB + PSRSRR | -0.005 | **0.079** | **170.091** |

Therefore, by detailed theoretical derivation and simulation study, we have shown that our proposed algorithm PSRSRR is a general acceleration algorithm that could be combined with many existing advanced methods using distance metric besides the basic Mahalanobis distance.

## B.6 COMPARISON OF COMPUTATIONAL COST BETWEEN ALGORITHM 2 AND 3

In Algorithm 2, the first step is that the chain would run till the max iteration number $N$ to generate a candidate assignment, and then in the second step, this assignment is determined to be accepted or rejected based on certain acceptance probability, and this above procedure would be repeated until

the required number of assignments has been collected. While in algorithm 3, an early stopping in the chain evolution step is applied in order to shorten the sampling time while aiming to still approximate well enough the stationary distribution. Below, we conduct comparison between these two algorithms with different max iteration number to carefully examine their computational costs.

We continue to use the simulation settings as described in Section 4.1. For simplicity, we fix $R^2 = 0.5$, and use the non-effect scenarios, with different max iteration number $N$ and $(n, p)$ pairs. We generate 100 assignments for each setting using the same threshold $p_a = 10^{-3}$ (this is chosen relatively large to reduce overall computational costs while strictly ensuring comparability) and summarize the results in Table 7.

Table 7: MSE and sampling time comparison between Algorithm 2 under different max iteration numbers and Algorithm 3 with the proposed stopping rule.

| $(n, p)$ | $N$ | MSE | Sampling Time (s) | $(n, p)$ | $N$ | MSE | Sampling Time (s) |
|----------|-----|-----|-------------------|----------|-----|-----|-------------------|
| | 1000 | 0.031 | 869.954 | | 1000 | 0.022 | 1422.150 |
| (250,2) | 10000 | **0.024** | 37208.290 | (500,2) | 10000 | **0.016** | 24924.690 |
| | PSRSRR | 0.027 | **5.858** | | PSRSRR | 0.016 | **11.293** |
| | 1000 | 0.107 | 782.018 | | 1000 | **0.036** | 1410.902 |
| (250,5) | 10000 | **0.088** | 8117.958 | (500,5) | 10000 | 0.047 | 13818.710 |
| | PSRSRR | 0.095 | **3.446** | | PSRSRR | 0.037 | **8.014** |

We have used relatively large max iteration number $N$ and moderate sample size $n$ for comparison. From Table 7, we can conclude that our algorithm has MSE values close to the algorithm using large $N$, while it has significantly smaller sampling times, as the algorithm with $N = 10^4$ can have sampling times of the order of $10^5$ seconds, indicating that simply setting a large $N$ to ensure that the chain has converged close enough to the stationary distribution would be almost computationally infeasible. Together, these findings support the effectiveness of our stopping rule.

### B.7 MORE DETAILS OF REAL DATA APPLICATIONS

**Analysis of Reserpine data.** In this dataset, the treatment group has 20 participants while the control group has 10. Similar to Zhu & Liu (2023), we include 8 important covariates to balance, and the threshold $a$ is set as the $p_a = 0.001$ quantile of $\chi_8^2$, i.e., $a = 0.86$ for all methods. Here, we exclude comparison with GSW as it could not have an exact treatment-control division as desired. And due to the small sample size, we also do not include the proposed method using $\nu_{p,a}$ for threshold selection, as this would only have correct interpretations in asymptotic scenarios.

We generate $10,000$ assignments, and obtain the following results. In Table 1, we compare the total time of generating these $10,000$ assignments, and in Figure 13, the empirical distributions of the standardized differences in each covariate mean are presented and the empirical percent reductions in variance (PRIVs) relative to CR are calculated. From these results, we can conclude that our proposed algorithm PSRSRR has achieved much faster sampling speed than the CR and much improved variance compared to the randomized design. Besides, the assignments generated using our proposed algorithm have a balance table for each covairate mostly within the recommended univariate balance thresholds $[-0.1, 0.1]$ (Austin, 2009) (illustrated with dashed lines).

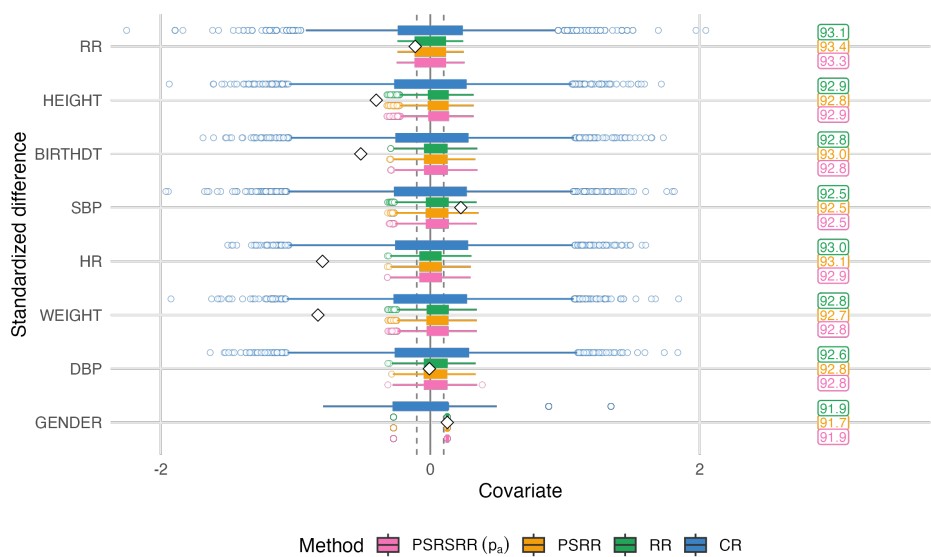

Figure 13: Box-plot of standardized differences in covariate means for Reserpine data. The diamonds indicates the standardized difference for the actual assignment in the experiment.

**STAR data in Angrist et al. (2009) and its preprocessing.** Student Achievement and Retention (STAR) Project is a randomized experimental evaluation of strategies designed to improve academic performance among college freshmen. Students in this experiment, except for those with a high school grade point average (GPA) in the upper quantile, were randomly assigned to one of three treatment groups or a control group. To keep the situation simple, similar to Li et al. (2018) and Wang & Li (2022), we keep only one treatment group, which received both additional mentoring services as well as incentives in the form of substantial cash awards for meeting a target GPA, to compare against the control group that was only eligible for standard university support services but nothing extra. And following their data preprocessing procedure, we discard students with missing data in the following covariates, or the first year GPA which is used as the observed outcome in their analysis, resulting in a treatment group of $n_1 = 118$ and control group of $n_0 = 856$. Considering the practical implications regarding the number of covariates given in Wang & Li (2022), we keep the first five covariates, i.e., high-school GPA, whether lives at home, gender, age and whether rarely puts off studying for tests, in our design stage.

Table 8: Covariates by tier in STAR data as in Li et al. (2018).

| Tier | Covariates |
|------|------------|
| Tier 1 | High-school GPA |
| Tier 2 | Whether lives at home, gender, age |
| | Whether rarely puts off studying for tests |
| Tier 3 | Whether mother/father is a college graduate |
| | Whether mother/father is a high-school graduate |
| | Whether never puts off studying for tests |
| | Whether wants more than a bachelor degree |
| | Whether intends to finish in 4 years |
| | Whether plans to work while in school |
| | Whether at the first choice school, mother tongue |

## C    EXTENDED RELATED WORK

Randomized experiments and A/B tests are foundational tools in causal machine learning and treatment effect estimation (Cai et al., 2024; Wang et al., 2023a; Byambadalai et al., 2025). As an experimental design strategy, rerandomization (RR) improves covariate balance, thereby reducing variance and enhancing the efficiency of causal estimators. For an accessible introduction, we refer readers to Section 6.1 of Ding (2024), while Li et al. (2018) and Wang & Li (2022) provide rigorous asymptotic analyses.

RR has been extended to a broad range of experimental settings, including factorial designs (Branson et al., 2016; Li et al., 2020), sequential designs (Zhou et al., 2018), stratified experiments (Johansson & Schultzberg, 2022; Wang et al., 2023b; Wang & Li, 2024; Cytrynbaum, 2024), matched-pair designs (Kalbfleisch & Xu, 2023), cluster experiments (Lu et al., 2023), interference settings (Basse & Airoldi, 2018; Zhang, 2025), survey experiments (Yang et al., 2023b), split-plot designs (Shi et al., 2024), and response-adaptive designs (Zhang & Yin, 2021).

A rich collection of balance criteria has also been proposed, including tiers of covariates (Morgan & Rubin, 2015), weighted Mahalanobis distances (Lu et al., 2023), $p$-value–based criteria (Zhao & Ding, 2024), ridge criteria (Branson & Shao, 2021), Bayesian criteria (Liu et al., 2025), and general quadratic-form frameworks (Schindl & Branson, 2024). In high-dimensional settings, practitioners often reduce dimensionality using Principal Component Analysis (PCA) (Zhang et al., 2024) or select covariates using pre-experimental or longitudinal pilot data (Johansson & Schultzberg, 2020).

RR has also been applied in biomedical and social science studies (Maclure et al., 2006; Lee et al., 2021; Abaluck et al., 2022; Henderson & Han, 2022; Resnjanskij et al., 2024). Despite its broad adoption, a persistent challenge remains: classical rejection sampling becomes computationally prohibitive when the optimal acceptance probability is small. Our work addresses this limitation by developing a computationally efficient sampling framework that enables RR at theoretically optimal thresholds in Wang & Li (2022), thereby broadening its applicability in practical causal ML pipelines.

## D    LLM USAGE

We used ChatGPT-5 to improve the clarity of our writing by correcting grammar, refining sentence structure, and ensuring stylistic consistency.

We used Gemini 2.5 Pro and ChatGPT-5 to help structure the proofs and draft the theoretical explorations in Appendix A.3. All results and derivations were rigorously verified and finalized by the authors.

