# OpenReview forum: "Fast Rerandomization for Balancing Covariate in Randomized Experiments: A Metropolis–Hastings Framework"
_ICLR.cc/2026/Conference — Submitted to ICLR 2026_

### Official Review · Reviewer_yBgd · 2025-10-31

**Soundness:** 4
**Presentation:** 4
**Contribution:** 4
**Rating:** 6
**Confidence:** 2

**Summary:**

The paper addresses a well-known limitation of rerandomization in experimental design: the computational inefficiency of classical rejection sampling under small balance thresholds. The authors propose a Metropolis–Hastings–based framework to accelerate rerandomization while retaining theoretical guarantees of uniformity over acceptable treatment assignments.

**Strengths:**

1. Clear problem motivation: The paper identifies the genuine computational bottleneck of rerandomization.
2. Strong theoretical grounding: The paper rigorously proves stationarity and uniformity.
3. The method proposed looks very clear and practical for me.
4. The paper is well writtern and easy to follow.

**Weaknesses:**

To be clear, I am not an expert in this field. I generally enjoy reading the paper and am not fully confident in providing any weakness. Below are just what in my mind, which can be wrong.

1. Technical contribution: Although there are some interesting ideas in designing the algorithm, the analysis and some ideas themselves (including the rejection sampling) seem to be standard. The main contribution seems to be applying them to the rerandomization context.
2. More general discussion on how to set the temperature can be helpful, together with some sensitivity analysis of the parameter $T$.
3. The paper is quite statistical-methodological with limited ML relevance. I think it would be beneficial to frame it for a more general ICLR audience.

**Questions:**

See above.

---

> ### Author Response · Authors · 2025-11-23
>
> We sincerely thank the reviewer for carefully reading our paper and for the constructive comments.
>
> ### 1. Technical contribution
>
> As a first step toward a theoretical understanding of our practical algorithm, we have added analysis of a generalized framework in the Appendix. This framework modifies our algorithm by enforcing two explicit constraints: a burn-in period $L_{burn}$, during which the chain evolves without performing rejection sampling to ensure convergence to stationarity; and a sampling interval $s_{chk}$, such that the rejection sampling step is performed only every $s_{chk}$ steps. We prove that if the burn-in is sufficiently long and the rejection sampling steps are sufficiently spaced, the output distribution of this generalized process converges to the target uniform distribution in total variation distance.
>
> Our algorithm corresponds to the limiting case of this framework where $L_{burn}=0$ and $s_{chk}=1$. While we do not provide a formal non-asymptotic bound for this specific parameter setting, we hypothesize that the algorithm relies on an ''implicit burn-in'' mechanism: since the acceptance probability is typically very small, the algorithm naturally runs for a large number of iterations before accepting a candidate. If this expected waiting time significantly exceeds the mixing time of the Markov chain, the distribution is expected to converge to stationarity before acceptance occurs, thereby mimicking the behavior of the explicit burn-in process. We leave the rigorous characterization of this implicit burn-in regime as a direction for future work.
>
> ### 2. Temperature strategy and sensitivity analysis of $T$
> We have added a sensitivity analysis of temperature $T$ in **Appendix B.2** by using different temperature values in different settings, and summarized the comparison regarding MSE in **Figure 4**, and the comparison regarding sampling time in **Figure 5**.
>
> In terms of MSE, we can conclude that $T$ has no significant influence, with smaller $T$ performing slightly better in small-sample, high-dimensional settings (e.g. $n=250$ and $p \in\{10,25\}$ ). As for sampling time, **Figure 5** shows that very small temperature (especially $T=0.01$) can dramatically increase the sampling time when $n$ is large, while this increase has the tendency to decrease when $p$ gets larger with fixed $n$ (note the time axis differs in each row). And it is also noticeable that the optimal $T$ moves to $0$ when $p$ increases.
>
> Therefore, combined with the above observation, our empirical choice $T=1.8/p$ lands in a stable and favorable position with these trends under different $(n,p)$, which effectively moves to smaller values when $p$ increases, resulting in better MSE and shorter sampling times. This choice also reflects the intuition that the increase in $p$ leads to smoothness in the Mahalanobis distance, indicating a smaller change in $M(\mathbf{W})$ after a single pair-switch and it is thus reasonable for the algorithm to favor less exploration.

---

> > ### Author Response · Authors · 2025-11-23
> >
> > ### 3. ML relevance
> >
> > We have expanded the discussion in the related work (**Appendix C**) to clarify the relevance to the broader ML community. Carefully designed randomized experiments play a central role in causal ML, A/B testing, and treatment effect estimation (Cai et al., 2023; Wang et al., 2023; Kato et al., 2024; Byambadalai et al., 2025). Rerandomization provides a principled way to improve covariate balance, which in turn reduces variance and yields more efficient causal estimators. This has been demonstrated in settings such as network interference (Zhang, 2025), cluster-level experiments (Lu et al., 2022), and survey or service experiments (Yang et al., 2021). We refer readers to our extended related work section for additional examples.
> >
> > A key challenge in applying rerandomization in these ML-relevant scenarios is the computational inefficiency of classical rejection sampling, especially when the optimal acceptance threshold is small. Our paper addresses this limitation by providing a computationally efficient sampling framework that enables rerandomization with theoretically optimal thresholds, thereby making rerandomization more broadly applicable in practical causal ML workflows.
> >
> > ### Reference
> >
> > Cai, C., Zhang, X., and Airoldi, E. (2023). *Independent-Set Design of Experiments for Estimating Treatment and Spillover Effects under Network Interference.* ICLR.
> >
> > Wang, J., Li, P., and Hu, F. (2023). *A/B testing in network data with covariate-adaptive randomization.* ICML.
> >
> > Byambadalai, U., Hirata, T., Oka, T., and Yasui, S. (2025). *On Efficient Estimation of Distributional Treatment Effects under Covariate-Adaptive Randomization.* ICML.
> >
> > Kato, M., Oga, A., Komatsubara, W., and Inokuchi, R. (2024). *Active Adaptive Experimental Design for Treatment Effect Estimation with Covariate Choice.* ICML.
> >
> > Zhang, Q. (2025). *Rerandomization Algorithms for Optimal Designs of Network A/B Tests.* Technometrics.
> >
> > Lu, X., Liu, T., Liu, H., and Ding, P. (2022). *Design-Based Theory for Cluster Rerandomization.* Biometrika.
> >
> > Yang, Z., Qu, T., and Li, X. (2021). *Rejective Sampling, Rerandomization and Regression Adjustment in Survey Experiments.* JASA.

---

> > > ### Comment · Reviewer_yBgd · 2025-11-27
> > >
> > > Thank you very much for your careful response. As I have mentioned, I am not an expert in this field. I am comfortable to raise the score to 7 if there is such a choice.

---

> > > > ### Author Response · Authors · 2025-11-27
> > > >
> > > > Thank you very much for your positive feedback and for your willingness to revise your assessment. We truly value your comments, as input from generalists helps us improve the clarity and broader impact of the paper.

---

### Official Review · Reviewer_phQ3 · 2025-10-31

**Soundness:** 3
**Presentation:** 4
**Contribution:** 2
**Rating:** 4
**Confidence:** 3

**Summary:**

This paper proposes PSRSRR, a Metropolis–Hastings–based framework for ensuring uniformity when accelerating rerandomization in randomized experiments. Classical rerandomization acceleration method can ensure covariate balance but does not ensure uniformity over the feasible region, making classical theoretical inference results for rerandomization inapplicable. The authors model pair-switching of treatment assignments as a Markov chain, prove its stationary distribution, and then apply a sampling-importance resampling step to restore uniformity. A practical “early-stopping” version is introduced to trade exactness for speed. Simulation and real-data experiments suggest PSRSRR achieves similar statistical validity to classical rerandomization while being substantially faster.

The paper is methodologically careful and builds on solid theoretical ground. The derivations of the stationary distribution and uniformity correction are mathematically correct and appropriately referenced. The experimental setup is standard and sufficient to demonstrate feasibility.

However, the claims of theoretical novelty are somewhat overstated—the work mainly adapts standard MCMC and rejection-sampling concepts to rerandomization. The early-stopping heuristic, which drives most of the practical efficiency, lacks a rigorous analysis of its bias relative to the true uniform distribution.

The paper is clearly written, with a logical flow and helpful diagrams. The algorithms are well-specified. References are thorough and well-chosen.

**Strengths:**

-	Provides a clean Markov-chain interpretation of pair-switching rerandomization as well as additional rejection sampling with clear theoretical justification.
-	Demonstrates solid empirical performance: substantial computational gains with minimal loss of balance quality.
-	Algorithm is clear and easy to implement.

**Weaknesses:**

This paper proposes an algorithm that ensures uniformity when conducting pair-switching rerandomization, which is Algorithm 2 in the paper. However, Algorithm 2 introduces additional rejection sampling step, making the procedure even slower. Thus, the author proposes a heuristic approach that does rejection at every step, which is the PSRSRR in the paper. The main problem is that the heuristic PSRSRR lacks theoretical guarantee on uniformity with only empirical evidence shows improvement on uniformity over the PSRR algorithm.

For this reason, I am wondering if there’s a theoretically rigorous way for conducting inference given the PSRSRR sampling scheme. For example, whether the inference procedure in (Zhu and Liu, 2022) is applicable here. On the other side, the author could also give some theoretical guarantee on the deviation from uniformity from the proposed procedure and explain how that deviation will influence the inference procedure.

To summarize:
-	Limited novelty: core ideas (pair-switching + MCMC + resampling) are straightforward extensions of prior PSRR and general MCMC principles.
-	Heuristic implementation lacks theoretical error bounds or diagnostic tools to quantify deviation from uniformity.  Only numerical evidence of improvement is shown.

**Questions:**

See above.

---

> ### Author Response · Authors · 2025-11-23
>
> We are very grateful for the reviewer's careful review and constructive comments.
>
> ### 1. Theoretical error bound for the PSRSRR
>
> As a first step toward a theoretical understanding of our practical algorithm, we have added analysis of a generalized framework in the Appendix. This framework modifies our algorithm by enforcing two explicit constraints: a burn-in period $L_{burn}$, during which the chain evolves without performing rejection sampling to ensure convergence to stationarity; and a sampling interval $s_{chk}$, such that the rejection sampling step is performed only every $s_{chk}$ steps. We prove that if the burn-in is sufficiently long and the rejection sampling steps are sufficiently spaced, the output distribution of this generalized process converges to the target uniform distribution in total variation distance.
>
> Our algorithm corresponds to the limiting case of this framework where $L_{burn}=0$ and $s_{chk}=1$. While we do not provide a formal non-asymptotic bound for this specific parameter setting, we hypothesize that the algorithm relies on an ''implicit burn-in'' mechanism: since the acceptance probability is typically very small, the algorithm naturally runs for a large number of iterations before accepting a candidate. If this expected waiting time significantly exceeds the mixing time of the Markov chain, the distribution is expected to converge to stationarity before acceptance occurs, thereby mimicking the behavior of the explicit burn-in process. We leave the rigorous characterization of this implicit burn-in regime as a direction for future work.
>
> ### 2. Theoretically rigorous inference procedure
>
> The Fisher randomization test (FRT) used in Zhu and Liu (2022) remains exactly valid under PSRSRR, as long as the same assignment mechanism is used both when generating the treatment assignment and when constructing the reference distribution for FRT.
>
> For asymptotic inference, uniformity of the assignment distribution is required. This, in turn, requires that the deviation from the target uniform distribution converges to zero, which will be ensured by the theoretical bound discussed above. Once this deviation vanishes, the standard asymptotic theory for rerandomization applies.
>
> ### 3. Novelty
>
> To the best of our knowledge, our paper is the first to provide a computationally efficient sampling algorithm that enables the use of the optimal diminishing acceptance probability $p_a$, which is essential for achieving statistically optimal designs. This makes it possible to apply both valid Fisher randomization tests and valid asymptotic inference within the same rerandomization framework.

---

> > ### Author Response · Authors · 2025-11-23
> >
> > ### 4. Diagnostic tools to quantify deviation from uniformity
> >
> > In **Appendix B.3**, we have included our proposed uniformity evaluation methods.
> >
> > One in **Appendix B.3.1** uses the treatment assignments generated by RR (classical rerandomization) as baseline, and conducts a Kolmogorov-Smirnov test to check if Mahalanobis distance distribution of this baseline and that of the assignments generated by our proposed algorithm PSRSRR are the same. Since RR is computationally expensive in large sample and small $p_a$ settings, we mainly test the uniformity in small sample cases with comparison with PSRR. The boxplots of p-values of the Kolmogorov-Smirnov test verify that PSRSRR can generally be considered as a good approximation of RR in terms of Mahalanobis distance as the nulls are not rejected in these settings, and PSRSRR is also a better approximation than PSRR.
> >
> > The other one in **Appendix B.3.2** directly uses the theoretical result in Morgan & Rubin (2012) that the asymptotic acceptance probability of the treatment assignment is $p_a={\mathbb P}(\chi_p^2\leq a)$, and therefore the asymptotic distribution of the Mahalanobis distances of assignments sampled uniformly from the acceptance space is a truncated chi-square distribution $M_{\rm asym}\sim\chi_p^2|\chi_p^2\leq a$. We generate this truncated chi-square as baseline, and conduc a Kolmogorov-Smirnov test to check if the Mahalanobis distribution of the assignments generated by our proposed algorithm PSRSRR follows this theoretical asymptotic distribution. Since this diagnostic method is much computational efficient, we extend the checks into larger sample size scenarios and choose $p_a$ according to the $\nu_{p,a}$ criteria as we have discussed in Section 4.1 following the practial implications given in Wang and Li (2022). With sample size large as $n=2000$ and threshold as small as $p_a = 10^{-20}$, we have shown that the nulls would not be rejected in many different settings, thus verifying that the assignments generated by our proposed algorithm PSRSRR can be as good as uniform in terms of Mahalanobis distance.
> >
> > #### Reference
> > Ke Zhu and Hanzhong Liu. (2022). *Pair-switching rerandomization.* Biometrics.
> >
> > Kari Lock Morgan and Donald B Rubin. (2012). *Rerandomization to improve covariate balance in experiments*. The Annals of Statistics.
> >
> > Yuhao Wang and Xinran Li. (2022). *Rerandomization with Diminishing Covariate Imbalance and Diverging Number of Covariates*. Annals of Statistics.

---

> > > ### Comment · Reviewer_phQ3 · 2025-11-27
> > >
> > > Thank you for the additional information. I will maintain my rating.

---

> > > > ### Author Response · Authors · 2025-11-27
> > > >
> > > > Thank you very much for your time and for carefully reviewing our paper, as well as for considering our additional information. We appreciate your feedback and have revised the paper accordingly. We are happy to provide any further clarification if needed.

---

### Official Review · Reviewer_nhiY · 2025-10-31

**Soundness:** 3
**Presentation:** 2
**Contribution:** 3
**Rating:** 6
**Confidence:** 3

**Summary:**

Rerandomization aims at producing a random treatment assignment that yields treatment and control groups with similar covariate means, measured by the Mahalanobis distance. For some theoretical results and methods to apply, it is useful to sample a treatment assignment uniformly from the set of all assignments whose Mahalanobis distance between groups does not exceed a small threshold $a > 0$. However, existing rerandomization methods either fail to guarantee uniformity or lack computational efficiency.

First, this paper introduces a rerandomization algorithm called Truncated Pair-Switching (Algorithm 1). The authors derive the explicit form of the limiting and stationary distribution (Theorem 1) of the Markov chain generated by this algorithm.

Then, they propose Algorithm 2, which samples exactly from the desired uniform distribution (Theorem 2) by applying rejection sampling to Algorithm 1, where the acceptance probability is inversely proportional to the limiting probability obtained in Theorem 1, truncated to the target set of assignments.

As a computationally cheaper heuristic, the authors integrate Algorithms 1 and 2 into a single loop rather than nesting them, forming Algorithm 3 (Pair-Switching Rejection Sampling Rerandomization; PSRSRR).

Experiments on both synthetic and real datasets compare the proposed method (PSRSRR) with existing ones in terms of mean squared error (MSE), confidence interval (CI) length, coverage rate, statistical power, type I error, and computational time. The proposed method consistently performs among the best in both statistical accuracy and computational efficiency, with especially strong advantages in higher-dimensional settings.
The paper also confirm that the PSRSRR produces uniform samples despite being a heuristic.

**Strengths:**

- The paper provides useful theoretical guarantees (Theorem 1 and Theorem 2), which seems to be a novel and good contribution to the literature.

- The proofs of Theorems 1 and 2 seem correct to me (except the part using Lemma 4, for which the notation is not clear).

- The heuristic version of the algorithm (Algorithm 3) is computationally efficient (Figure 2), while being superior or comparable to previous methods in terms of statistical performance (Figure 1).

- The proofs are also written in an accessible way.

**Weaknesses:**

- Algorithms 1 and 2 are similar to PSRR (Zhu and Liu, 2022) especially when we set $N = 1$, but the paper does not clearly mention this fact.

- I appreciate that the authors' effort to make the proof self-contained by restating the theorem from Harchol-Balter (2024) as Lemma 5. However, there are undefined symbols and I do not understand what it states.

- There are a few unclear points in the proof of Theorem 1, which is making it difficult to understand. (See the Questions.)

- PSRR is slightly faster for larger $p$ in Figure 2.

- There seems to be no numerical results for Algorithm 2.

- It is difficult to see if the proposed method is better than GSW. GSW seems advantageous in the coverage rate while PSRSRR is better in CI length, but these criteria might be in a trade-off relationship.

**Questions:**

### Major comments
- Please clearly explain the contribution relative to Zhu and Liu (2021).

- Please rephrase Lemma 5 using symbols defined in this paper.

- In Theorem 2 and the text below it, shouldn't the part of PSRSRR be rather Algorithm 2? I thought PSRSRR refers to Algorithm 3, which has no theoretical guarantee.

-----
### Minor comments
- If we can interpret Algorithm 1 as a Metropolis-Hastings method, I thought there is a proposal distribution. What is the proposal distribution?

- l.64: "robust alternative"---alternative to which method?

- l.122: what is $Cov(...)$ exactly?

- In Corollary 3, what is "the projection of potential outcomes on covariates" exactly?

- In Corollary 3, $S_X^2$ --> $S_{XX}$?

- In A.2 PROOF OF THEOREM 2, $g$ might not be defined.

- Are there any numerical results for Algorithm 2? In particular, it would be interesting to compare its running times with those of Algorithm 3.

- ll.321-323: "We provide strong indirect evidence for the near-uniformity"---Where can we find this result?

- Around l.698 in the proof of Theorem 1, we might need $n_c + n_t \ge 3$ to be able to find such three indices $p, s, q$.

---

> ### Author Response · Authors · 2025-11-23
>
> We sincerely thank the reviewer for a thorough and thoughtful review, with comments that are both in-depth and attentive to notational and presentational details. We have revised all points in the paper and respond point by point as follows. We use “W” for Weakness, “Ma” for Major comments, and “Mi” for Minor comments, referring to the numbering in the reviewer’s report.
>
> ### W1 + W4 + Ma1. Contribution relative to Zhu and Liu (2022)
>
> Zhu and Liu (2022) proposed PSRR to accelerate rerandomization (RR), but it targets a **different assignment distribution**. PSRR does not generate assignments uniformly from the acceptable set and therefore cannot inherit the existing asymptotic theory established for RR. It mainly relies on Fisher randomization tests for inference.
>
> In contrast, Algorithm 1 in our paper aims to draw assignments from the stationary distribution induced by the pair-switching Markov chain, and Algorithm 2 applies rejection sampling on top of this stationary distribution. This key idea allows us to **recover the uniform distribution** over all acceptable assignments, which is the target of RR. As a result, our method improves computational efficiency while inheriting all existing asymptotic theory of RR.
>
> We also acknowledge that PSRR can be slightly faster than PSRSRR in some settings, since PSRSRR pays a computational cost to match RR’s target distribution.
>
> ### W2 + Ma2. Notation in Lemma 5
>
> We have simplified Lemma 5 **(now Lemma 4)** to only include results that are necessary for our proof to improve readibility and clearity.
>
> ### Ma3. Clarifying that Theorem 2 corresponds to Algorithm 2
>
> The text below Theorem 2 has been corrected to explicitly refer to Algorithm 2, and PSRSRR does refer to Algorithm 3.
>
> ### W3. Clarifying the proof of Theorem 1
>
> We have reorganized and clarified several steps in the proof of Theorem 1 to improve readability. In the current **Appendix A.1**, we now include a rigorous formulation of the pair-switching Markov chain, and explicitly prove Theorem 1 in two steps, first by verifying the proposed distribution $\pi$ is the stationary distribution via detailed balanced condition, and second by showing the chain satisfies irreducibility and aperiodicity to invoke Lemma 4 to conclude the proof.
>
> ### W5 + Mi7. Computational comparison between Algorithm 2 and Algorithm 3
>
> We have added experiments to compare the MSE and sampling time of these two algorithms under different settings and with relatively large max iteration number $N$ in Algorithm 2 to approximate a sampling from the stationary distribution. The results are summarized in **Table 7 in Appendix B.6**. From the results, we can conclude that our algorithm has MSE values close to the algorithm using large $N$, while it has significantly smaller sampling times, as the algorithm with $N=10^4$ can have sampling times of the order of $10^5$ seconds, indicating that simply setting a large $N$ to ensure that the chain has converged close enough to the stationary distribution would be almost computationally infeasible. Together, these findings support the effectiveness of our stopping rule.
>
> ### W6. Comparison with GSW
>
> In our simulations, GSW produces conservative intervals with coverage above the nominal level, which is also reported in Harshaw et al. (2024). Harshaw et al. (2024) state that RR with a diminishing threshold achieves comparable limiting variance to GSW but faces computational challenges. Our paper directly addresses this computational issue by making RR practically scalable.
>
> Moreover, RR has advantages of (i) conceptual simplicity, (ii) applicability across a wide range of experimental designs, and (iii) well-developed theoretical support, as reviewed in the related work section.

---

> ### Author Response · Authors · 2025-11-23
>
> ### Mi1. Proposal distribution in Algorithm 1
>
> Given the current assignment $\mathbf{W}^{(t)}$, let $\mathscr{W}^{(t)}$ denote the set of assignments obtained by randomly switching one treated unit and one control unit in $\mathbf{W}^{(t)}$. The proposal distribution is the uniform distribution over $\mathscr{W}^{(t)}$.
>
> ### Mi2. “Robust alternative”
>
> We clarified that Fisher randomization tests have been widely advocated as a robust alternative to **asymptotic inference (Li et al., 2018)**.
>
> ### Mi3: l.122 (now l.132) What is $Cov(\dots)$ exactly?
>
> We now explicitly write out the expression
> $\operatorname{Cov}(\bar X\_t - \bar X\_c)
>   = \dfrac{n}{n\_t n\_c} S\_{XX}$ in **Section 2.2**
> when giving the definition of the Mahalanobis distance, and
> $S\_{XX}
>   = \dfrac{1}{n-1} \sum\_{i=1}^{n}
>     (X\_i - \bar X)(X\_i - \bar X)^\top$,
> which has been given before in **Section 2.1**.
>
>
>
> ### Mi4: In Corollary 3, what is "the projection of potential outcomes on covariates" exactly?
> We have added its mathematical formula in **Corollary 3** for clarity:
> $s\_{Y(i)\mid X}^2
>   = s\_{Y(i),X} S\_{XX}^{-1} s\_{X,Y(i)}$.
>
>
> ### Mi5: In Corollary 3, $S_X^2$-->$S_{XX}$?
> We have corrected $S_X^2$ to $S_{XX}$ to improve consistency and readability.
>
>
> ### Mi6: In A.2 PROOF OF THEOREM 2, $g$ might not be defined.
> We have corrected our notations from the undefined $g$ to the stationary distribution $\pi$, as that would be the distribution used in this sampling procedure in our case.
>
>
> ### Mi8: l.321-323 (now l.357-358): "We provide strong indirect evidence for the near-uniformity"---Where can we find this result?
> We indirectly verify the near-uniformity of the assignment generated by our proposed algorithm in **Section 4.1** and **Appendix B.4** when showing its performance in coverage rate and Type I error. Since we are using theoretical result in Corollary 3, which is originally derived for assignments sampled uniformly from the acceptance space, to construct confidence interval now using our assignments, and the simulation studies show that the coverage rate and Type I error constructed this way could still be satisfactory (coverage rate close to 95% and Type I error approximately below 0.05). We interpret this as indirect evidence of the near-uniformity property of our assignments, since if the assignments are not uniform enough, then the inference results using Corollary 3 would not have expected performance as if using assignments sampled uniformly from the acceptance space.
>
> ### Mi9: Around l.698 (now l.821) in the proof of Theorem 1, we might need $n_c+n_t \geq 3$ to be able to find such three indices $p, s, q$.
>
> We have added the assumption that $n_t\geq 2, n_c\geq 2$ in **Section 2.1**.
>
>
>
> ### Reference
> Ke Zhu and Hanzhong Liu. (2022). *Pair-switching rerandomization.* Biometrics.
>
> Christopher Harshaw, Fredrik Sävje, Daniel A Spielman, and Peng Zhang. (2024). *Balancing covariates in randomized experiments with the gram–schmidt walk design.* Journal of the American Statistical Association.
>
> Xinran Li, Peng Ding, and Donald B Rubin. (2018). *Asymptotic theory of rerandomization in treatment–control experiments.* Proceedings of the National Academy of Sciences.

---

### Official Review · Reviewer_VL23 · 2025-11-01

**Soundness:** 4
**Presentation:** 4
**Contribution:** 3
**Rating:** 8
**Confidence:** 5

**Summary:**

The paper proposed a new sampling method to generate a uniform distribution on all acceptable assignments in the context of randomized experiments. It generalizes existing methods based on pair-switching to a two-step reject sampling algorithm with uniformity guarantee and further heuristically combines the two steps into a single-chain continuously reject sampling algorithm. The proposed method is proven to be efficient and accurate with high sampling quality through simulation studies.

**Strengths:**

- **originality**: the proposed method is novel and effective. It adds to the existing toolbox of randomization algorithms.
- **quality**: the paper includes theoretical results along with extensive solid simulations.
- **clarity**: the paper is clearly drafted.
- **significance**: The efficient algorithm to generate high-quality random assignments to experimental units is an essential component in experimental design.

**Weaknesses:**

Although the paper provides a theoretical guarantee for the two-stage reject sampling algorithm, the practical algorithm is a heuristic revision of the theoretically justified one. The author empirically investigated the impact of the ad-hoc adaption, but there is still a theoretical gap. It would be perfect if any theoretical guarantee for the single-chain algorithm could be developed.

**Questions:**

- The proof of Theorem 1 could be significantly simplified by (i) verify the proposed distribution is stationary (ii) irreducibility + aperiodicity of a finite MC implies uniqueness, existence and convergence of the stationary distribution.
- In figure 2, it looks like PSRSRR performs bad on small n large p cases. What could be the reasons?

---

> ### Author Response · Authors · 2025-11-23
>
> We appreciate the thoughtful comments and constructive suggestions, and we address them below.
>
> ### 1. Theoretical guarantee for Algorithm 3
>
> As a first step toward a theoretical understanding of our practical algorithm, we have added analysis of a generalized framework in the **Appendix A.3**. This framework modifies our algorithm by enforcing two explicit constraints: a burn-in period $L_{burn}$, during which the chain evolves without performing rejection sampling to ensure convergence to stationarity; and a sampling interval $s_{chk}$, such that the rejection sampling step is performed only every $s_{chk}$ steps. We prove that if the burn-in is sufficiently long and the rejection sampling steps are sufficiently spaced, the output distribution of this generalized process converges to the target uniform distribution in total variation distance.
>
> Our algorithm corresponds to the limiting case of this framework where $L_{burn}=0$ and $s_{chk}=1$. While we do not provide a formal non-asymptotic bound for this specific parameter setting, we hypothesize that the algorithm relies on an ''implicit burn-in'' mechanism: since the acceptance probability is typically very small, the algorithm naturally runs for a large number of iterations before accepting a candidate. If this expected waiting time significantly exceeds the mixing time of the Markov chain, the distribution is expected to converge to stationarity before acceptance occurs, thereby mimicking the behavior of the explicit burn-in process. We leave the rigorous characterization of this implicit burn-in regime as a direction for future work.
>
> ### 2. Proof of Theorem 1 can be simplified
>
> In the revision, we have streamlined the proof by following the reviewer’s recommended structure: (i) verifying stationarity of the proposed distribution, and (ii) using irreducibility and aperiodicity of the finite-state Markov chain to ensure uniqueness and convergence. Revised version of the proof can be found in **Appendix A.1**.
>
> ### 3. Performance of PSRSRR in small $n$ and large $p$ settings (Figure 2)
>
> When $p$ is large relative to $n$, achieving covariate balance becomes inherently more challenging because the imbalance metric depends on many covariates. In this setting, small local updates such as pair switches may not sufficiently adjust the imbalance metric toward the acceptance region, which increases the number of iterations needed to reach an acceptable assignment.
>
> To improve performance under these scenarios, we have developed an adaptive temperature strategy that helps the sampler explore the assignment space more effectively and improves the acceptance behavior.
>
> In addition, when $p$ is very large, we recommend applying unsupervised (Zhang et al., 2023) or supervised (Johansson and Schultzberg, 2020) dimension reduction before rerandomization. These approaches typically improve both computational efficiency and statistical performance.
>
> ### Reference
>
> Zhang, et al. (2023) PCA Rerandomization. *Canadian Journal of Statistics*.
>
> Johansson and Schultzberg. (2020) Rerandomization Strategies for Balancing Covariates Using Pre-Experimental Longitudinal Data. *JCGS*.

---

### Official Review · Reviewer_x9ps · 2025-11-05

**Soundness:** 1
**Presentation:** 2
**Contribution:** 1
**Rating:** 2
**Confidence:** 4

**Summary:**

The paper considers a method for randomized experiments under a Metropolis-Hastings framework. The proposed algorithm efficiently balances covariates under experiments. The algorithm's performance is examined by simulations and two real-data applications.

**Strengths:**

This paper uses a well-known algorithm for covariate-balanced experiments and the method is immediately applicable.

**Weaknesses:**

- While the pair-switching strategy considered in the paper is noted as a proposal to reduce the computational cost, its primary limitation is that it functions merely as a algorithmic fix. It only addresses the speed of the re-randomization, not the fundamental criteria being optimized. Therefore, it it remains a "sub-optimal" design. It does not solve the main problem, which is the failure to prioritize covariates that are more strongly associated with potential outcomes. While the authors prioritize the uniform assignment over covariates, that should not be the objective.  Thus, the strategy in that paper is ultimately an "incomplete solution." By focusing narrowly on reducing the heavy computational burden, it fails to address the more significant, underlying theoretical deficiency. It offers no progress toward the more precise quantification of covariate heterogeneity that is needed. It simply provides a faster way to arrive at a sub-optimal result.

- The theoretical results presented in the paper appear to be only minor modifications of those in the existing literature. To strengthen the paper's contribution, the authors are encouraged to consult the paper cited below and other recent theoretical articles.

- Liu, Z., Han, T., Rubin, D. B., & Deng, K. "Bayesian Criterion for Re-randomization." arXiv preprint arXiv:2303.07904 (2023) and accepted at *Journal of the American Statistical Association*.

**Questions:**

Please refer to the limitations listed in *Weaknesses*.

---

> ### Author Response · Authors · 2025-11-23
>
> We appreciate the reviewer’s careful reading and constructive comments. Our responses are organized into four points.
>
> ### 1. Optimal design depends on both covariate importance (Liu et al., 2025) and the acceptance probability (Wang and Li, 2022).
>
> We fully agree that the goal should be an optimal rerandomization design. Such a design should (i) construct the imbalance metric using covariate importance weights that reflect associations with potential outcomes (Liu et al., 2025), and (ii) select an acceptance probability $p_a$ that decreases polynomially with the sample size $n$ (Wang and Li, 2022). Both components are necessary for achieving optimality. Even when using importance-weighted imbalance measures, choosing a large $p_a$ leads to a suboptimal design. For example, $p_a = 1$ is equivalent to complete randomization.
>
> Our work focuses on the second component, which is complementary to the imbalance metric emphasized by the reviewer. In the revision, we clarify this distinction and explain how our method can be combined with importance-weighted imbalance measures to obtain fully optimal designs.
>
> ### 2. Computational barriers lead to statistical suboptimality.
>
> Classical rejection sampling requires $1/p_a$ draws on average to obtain a single acceptable assignment. Because this may be computationally prohibitive for small $p_a$, practitioners often select fixed and suboptimal thresholds such as $p_a = 10^{-3}$ regardless of $n,p$, even though rerandomization with diminishing $p_a$ is theoretically optimal (Wang and Li, 2022).
>
> As a result, rerandomization often underperforms competing methods such as the Gram Schmidt Walk (Harshaw et al., 2024). The computational burden of rejection sampling is the main reason optimal $p_a$ is avoided. Our method removes this barrier, making it faster to use the optimal diminishing $p_a$ and recover the statistical optimality established by theory.
>
> ### **3. ReB can also be accelerated by our proposed algorithm.**
>
> We applied our acceleration strategy to rerandomization with the Bayesian criterion (ReB, Liu et al., 2025). Our experiment results summarized in the table below show two findings: ReB improves estimation efficiency relative to ReM, and our algorithm substantially reduces the runtime of ReB. This demonstrates that our acceleration framework is compatible with rerandomization criteria beyond Mahalanobis distance. More details regarding theoretical derivation and simulation studies could be found in **Appendix B.5**.
>
> **Table 6:** Bias, MSE and sampling time comparison between RR, ReB and ReB + PSRSRR under sample size $n \in \{100, 200, 500, 1000\}$.
>
> | Sample size \(n\) | Method         | Bias    | MSE    | Time (s) |
> |-------------------|----------------|--------:|-------:|---------:|
> | 100               | RR            | 0.064   | 1.378  | 173.143  |
> | 100               | ReB            | 0.058   | 1.311  | 164.481  |
> | 100               | ReB + PSRSRR | **0.003** | **1.015** | **39.601** |
> | 200               | RR       | **0.003** | 0.542  | 344.730  |
> | 200               | ReB            | 0.004   | 0.529  | 532.278  |
> | 200               | ReB + PSRSRR | -0.005  | **0.421** | **46.219** |
> | 500               | RR            | -0.015  | 0.268  | 894.383  |
> | 500               | ReB        | **-0.010** | 0.253  | 912.444  |
> | 500               | ReB + PSRSRR | 0.021   | **0.175** | **121.343** |
> | 1000              | RR       | **-0.002** | 0.112  | 1873.277 |
> | 1000              | ReB            | 0.002   | 0.097  | 1875.811 |
> | 1000              | ReB + PSRSRR | -0.005  | **0.079** | **170.091** |
>
>
>
> ### 4. Comprehensive literature review and contribution.
> We added an extended related-work section in **Appendix C** that summarizes methodological, theoretical, and applied advances in rerandomization. To the best of our knowledge, our paper is the first to provide a computationally efficient sampling algorithm that enables the use of the optimal diminishing acceptance probability $p_a$, which is essential for achieving statistically optimal designs.
>
> ### References
>
> Liu et al., 2025. A Bayesian Criterion for Rerandomization. *JASA*.
>
> Wang and Li, 2022. Rerandomization with Diminishing Covariate Imbalance and Diverging Number of Covariates. *Annals of Statistics*.
>
> Harshaw et al., 2024. Balancing covariates in randomized experiments using the Gram-Schmidt Walk. *JASA*.

---

### Author Response · Authors · 2025-12-03

We truly appreciate the Area Chair's extended time for deliberation due to the system issue and are grateful for the thoughtful reviews. Our core innovation addresses the most significant practical barrier to applying rerandomization designs: the extreme computational cost of exact rejection sampling.

Our algorithm, PSRSRR, provides a highly efficient, theoretically grounded sampling method that makes rerandomization practically usable at scale.


Below we summarize how our response and revisions address the main concerns.

---

### 1. Addressing the "Suboptimal Design" Critique (Reviewer x9ps)

Reviewer x9ps argued that focusing narrowly on speed is insufficient if the underlying design criterion (e.g., Mahalanobis distance) is suboptimal compared to methods that prioritize more important covariates.

Our framework is **not restricted to the Mahalanobis distance**. Instead, PSRSRR is a general, plug-and-play acceleration module that can be combined with alternative methods based on other covariate imbalance metrics, including those that emphasize more important covariates.

We demonstrate this using the Bayesian Criterion (ReB, Liu et al., 2025) as a primary case study in Appendix B.5. ReB incorporates knowledge on covariate importance by encoding it via a prior distribution. Our new experimental results (Table 6) show that:

* Superior Statistical Performance: ReB + PSRSRR consistently attains the lowest MSE across all tested sample sizes, confirming our framework enables statistically superior designs even with advanced criteria.

* Massive Speedup: ReB + PSRSRR reduces the sampling time of ReB by a factor of 4~12, effectively removing the computational barrier to using such advanced criteria.

---

### 2. Theoretical Grounding and Near-Uniformity of PSRSRR (Reviewer VL23, phQ3)

Reviewers correctly noted that PSRSRR (Algorithm 3), while efficient, relies on an early-stopping heuristic and does not guarantee the exact uniform sampling as Algorithm 2 does. This distinction between the theoretical ideal and the efficient approximation was openly discussed as a direction for future work in the original submission, and we have now extensively reinforced the justification with additional theoretical exploration and empirical validation in the revised manuscript.

* **Theoretical Foundation:** Our approach is founded on two algorithms with rigorous theory:
    * Algorithm 1 shows that the pair-switching procedure defines a Markov chain and derives its stationary distribution. (Theorem 1).
    * Algorithm 2 uses this stationary distribution with rejection sampling to exactly recover $Unif(W_a)$, the uniform distribution over acceptable assignments (Theorem 2).

* **Theoretical Motivation for Approximation:** Algorithm 2 is computationally impractical (Table 7, Appendix B.6). PSRSRR is introduced as an efficient approximation. We motivate this approximation with a rigorous theoretical analysis (Appendix A.3) of a generalized framework that proves convergence to $Unif(W_a)$ in total variation distance when explicit burn-in ($L_{burn}$) and sufficient sampling intervals ($s_{chk}$) are used. While this analysis requires large values for $L_{burn}$ and $s_{chk}$, and PSRSRR corresponds to the limiting case with $L_{burn}=0$ and $s_{chk}=1$, we hypothesize that the efficacy of PSRSRR arises from an "implicit burn-in" period due to the low acceptance probability.

* **Empirical Justification for Approximation:** 2 is computationally infeasible (sampling times on the order of $10^4$ seconds), whereas the theoretically motivated PSRSRR achieves comparable statistical quality (MSE) with a speed advantage of several orders of magnitude (on the order of only a few seconds).

* **Empirical Validation of Uniformity:**
    * **Direct** testing against the asymptotic truncated $\chi^2$ distribution in large-sample settings (Appendix B.3) yielded high p-values, indicating good agreement with the target distribution.
    * We also **indirectly** verify the near-uniformity of the assignment generated by our proposed algorithm in Section 4.1 and Appendix B.4 through its performance in statistical inference. The satisfactory performance of PSRSRR on metrics like Type I error (below $\alpha$) and confidence interval coverage rates (close to $1-\alpha$) provides strong indirect evidence that the generated assignments are distributed close to the target uniform distribution.

---

> ### Author Response · Authors · 2025-12-03
>
> ### 3. Comparative Performance: PSRSRR vs. GSW
>
> Reviewers compared PSRSRR with Gram-Schmidt Walk (GSW, Harshaw et al., 2024). Our results show the following advantages for PSRSRR:
>
> * **Statistical Efficiency:** PSRSRR consistently achieves a shorter CI length (up to $28$% shorter CI length, Table 4) and has MSE that is comparable to or modestly lower than GSW's (up to $13$% lower MSE, Table 3). Crucially, PSRSRR maintains coverage rates close to the nominal 95% level, whereas GSW's coverage is often noticeably above 95% (especially when $R^2$ is large; see Figure 12), indicating more conservative intervals. This confirms PSRSRR is more statistically efficient.
> * **Sampling Time:** On median, PSRSRR is dramatically faster than RR (over $1,800$ times faster) and is $4$ times faster than GSW (Figure 2).
>
> ---
>
> ### 4. Contribution Relative to Zhu and Liu (2022)
>
> Reviewers requested clarification on how our contribution relates to Zhu and Liu (2022).
>
> Zhu and Liu (2022) proposed PSRR to accelerate rerandomization (RR), but it targets a **different, non-uniform assignment distribution**. PSRR does not generate assignments uniformly from the acceptable set and therefore cannot inherit the existing asymptotic theory for RR. As a result, valid inference for PSRR typically relies on Fisher randomization tests, which **require repeatedly generating a large number of assignments** (often on the order of $10^3$~$10^4$).
>
> In contrast, our Algorithm 1 targets the stationary distribution of the pair-switching Markov chain; and Algorithm 2 applies rejection sampling to recover **the uniform distribution** over all acceptable assignments. As a result, our method inherits the existing asymptotic theory for RR, allowing standard asymptotic inference based on **a single realized assignment**, and yields more **efficient inference** in experiments (shorter confidence intervals and lower MSE; see Figure 1, Table 3~4) while inheriting all existing asymptotic theory of RR. We also empirically illustrate that PSRSRR yields assignments closer to uniform over the acceptance region than PSRR in Appendix B.3.1.
>
> ---
>
> ### Other Revisions
>
> * An expanded related work section in Appendix C covering methodological, theoretical, and applied advances in rerandomization.
> * Minor notation and writing improvements for clarity.
> * Hyperparameter temperature $T$ sensitivity analysis in Appendix B.2.
>
> ---
>
> ### Reference
>
> Zhaoyang Liu, Tingxuan Han, Donald B. Rubin, and Ke Deng. (2025). *A Bayesian criterion for rerandomization*. Journal of the American Statistical Association.
>
> Ke Zhu and Hanzhong Liu. (2022). *Pair-switching rerandomization.* Biometrics.
>
> Christopher Harshaw, Fredrik Sävje, Daniel A Spielman, and Peng Zhang. (2024). *Balancing covariates in randomized experiments with the gram–schmidt walk design.* Journal of the American Statistical Association.

---

### Meta-Review · Area_Chair_CsBp · 2026-01-07

**Summary:**

While the paper targets a genuine computational bottleneck in rerandomization, the methodological novelty is limited. The proposed Metropolis Hastings plus sampling importance resampling construction appears as a relatively straightforward combination of existing MCMC and correction techniques, and the extent to which it fundamentally differs from prior accelerated rerandomization or constrained randomization methods is not sufficiently clarified. In particular, the claim that uniformity is uniquely restored by the proposed procedure is not fully convincing, given the large literature on exact sampling from constrained assignment spaces.

The theoretical guarantees largely repackage known validity results for rerandomization, assuming that the sampler indeed targets the uniform distribution over acceptable assignments. However, the additional assumptions required for practical convergence of the Markov chain and the impact of finite time mixing are not carefully addressed. The empirical speedups, while impressive, are demonstrated in limited settings and do not fully account for tuning costs or scalability in high dimensional covariate spaces. Overall, the paper does not provide enough conceptual or theoretical advancement to warrant acceptance.

**Reviewer Concerns:**

Nothing to note

**Reviewer Scores:**

can't predict

---

### Decision · Program_Chairs · 2026-01-26

Reject